# Ventricular, atrial, and outflow tract heart progenitors arise from spatially and molecularly distinct regions of the primitive streak

**Kenzo Ivanovitch** [ORCID]*, **Pablo Soro-Barrio** [ORCID], **Probir Chakravarty** [ORCID], **Rebecca A. Jones** [ORCID], **Donald M. Bell** [ORCID], **S. Neda Mousavy Gharavy, Despina Stamataki, Julien Delile, James C. Smith, James Briscoe**

The Francis Crick Institute, London, United Kingdom

* kenzo.ivanovitch@crick.ac.uk

## Abstract

The heart develops from 2 sources of mesoderm progenitors, the first and second heart field (FHF and SHF). Using a single-cell transcriptomic assay combined with genetic lineage tracing and live imaging, we find the FHF and SHF are subdivided into distinct pools of progenitors in gastrulating mouse embryos at earlier stages than previously thought. Each subpopulation has a distinct origin in the primitive streak. The first progenitors to leave the primitive streak contribute to the left ventricle, shortly after right ventricle progenitor emigrate, followed by the outflow tract and atrial progenitors. Moreover, a subset of atrial progenitors are gradually incorporated in posterior locations of the FHF. Although cells allocated to the outflow tract and atrium leave the primitive streak at a similar stage, they arise from different regions. Outflow tract cells originate from distal locations in the primitive streak while atrial progenitors are positioned more proximally. Moreover, single-cell RNA sequencing demonstrates that the primitive streak cells contributing to the ventricles have a distinct molecular signature from those forming the outflow tract and atrium. We conclude that cardiac progenitors are prepatterned within the primitive streak and this prefigures their allocation to distinct anatomical structures of the heart. Together, our data provide a new molecular and spatial map of mammalian cardiac progenitors that will support future studies of heart development, function, and disease.

## Introduction

The cells that comprise different parts of an organ can arise from distinct origins and acquire their fate at different times during ontogeny. In the case of the heart, 2 mesodermal derived sources of cardiac progenitors, named the first heart field (FHF) and the second heart field (SHF), have been identified (reviewed in [1]). The FHF forms mainly the left ventricle and constitutes the initial cardiac crescent. By contrast, the cardiac progenitors of the SHF contribute to the right ventricle, outflow tract, and atria, in addition to branchiomeric muscles.

**Data Availability Statement:** Single cell RNA sequencing data have been deposited in NCBI under the accession number GSE153789.

**Funding:** K.I. has received funding from HFSP LTF (LT000609/2015-L). Work in the J.C.S. and J.B.'s lab was supported by the Francis Crick Institute, which receives its core funding from Cancer Research UK (FC001-157, FC001-051) the UK Medical Research Council (FC001-157, FC001-051), and the Welcome Trust (FC001-157, FC001-051). Work in the J.B.'s lab was also supported by the European Research Council under the European Union (EU) Horizon 2020 research and innovation program grant 742138. This research was funded in whole, or in part, by the Wellcome Trust (FC001-157, FC001-051). For the purpose of Open Access, the author has applied a CC BY public copyright license to any Author Accepted Manuscript version arising from this submission. The funders had no role in study design, data collection and analysis, decision to publish, or preparation of the manuscript.

**Competing interests:** The authors have declared that no competing interests exist.

**Abbreviations:** AHF, anterior heart field; AIP, anterior intestinal portal; aPS, anterior primitive streak; cTnnT, cardiac troponinin T; EB, early bud; EHF, early head fold; FBS, foetal bovine serum; FHF, first heart field; LB, late bud; nGFP, nuclear localised GFP; OB, no bud; pSHF, posterior second heart field; SA, surface area; SHF, second heart field; tam, tamofixen; UMAP, Uniform Manifold Approximation Projection; UMI, Unique Molecular Identifier.

Retrospective clonal analysis has suggested that the FHF and SHF progenitors segregate either before or at the onset of gastrulation [1,2]. The analysis showed large clones spanning multiple compartments. Clones belonging to the first lineage labelled the left ventricle and other compartments except the outflow tract, while the second lineage contributed to the outflow tract, atria, and right ventricle but never to the left ventricle. Subclones within the first and second lineages were also observed. These were smaller in size and suggested that the individualisation of the different regions of the heart happens at later stages [2].

Consistent with the notion that the cardiac lineages are established during gastrulation, clonal analysis based on tracing the progeny of *Mesp1*-expressing cells, which is expressed in the primitive streak and nascent mesoderm (https://marionilab.cruk.cam.ac.uk/MouseGastrulation2018/; [3–5]), indicated that independent sets of *Mesp1*-expressing cells contribute to FHF and SHF derivatives [6,7]. These can be distinguished by their time of appearance during gastrulation, with embryonic day (E)6.5 *Mesp1+* cells supplying the FHF while SHF derivatives preferentially derive from E7.5 *Mesp1+* cells [7]. The *Mesp1* clonal analysis resulted in clones of small sizes restricted to compartments derived from the FHF or SHF [6,7]. Thus, these observations are consistent with the existence of 2 groups of progenitors that subsequently became further restricted to specific cardiac fates in the mesoderm.

In addition to a separation between the FHF and SHF, genetic tracing experiments uncovered a distinction between the atria and ventricular progenitors. Genetic lineage tracing with the transcription factor *Foxa2* identifies a population of cardiac progenitors in the anterior primitive streak (aPS) at E6.5 that contribute to both left and right ventricles but not to the atria [8]. These results are in line with fate mapping studies in the chick, showing that the atrial and ventricular cells arise at different anterior–posterior positions in the primitive streak [9,10]. Although these analyses did not resolve the clonal relationship between the cells, these findings are compatible with a model in which sublineages within the FHF and SHF lineages already exist in the primitive streak, such that the atrial lineage is distinct from the right ventricle lineage within SHF progenitors. This would imply that 2 distinct cardiac progenitors exist solely at the epiblast stage, prior to gastrulation, and that more subpopulations of cardiac progenitors than first and second progenitors can be molecularly defined in the primitive streak. The most rigorous way to decide if this is the case is to define the location of all the cardiac progenitors in the streak and the precise embryonic stages at which they ingress. This would allow the comparison of distinct subpopulations of cardiac precursors and reveal putative molecular differences of cardiac progenitors in the primitive streak. Single-cell transcriptomic assays show *Mesp1+* cardiac progenitors diversify into molecularly distinct FHF and SHF populations at late embryonic day (E7.25) stage. This is once cells have migrated and reached their final location in the embryo [11]. The signalling environment cells encounter during and after their migration might play a role in the patterning of the cardiac progenitors into FHF and SHF domains [8,12–15]. Whether initial molecular differences between different sets of cardiac progenitors already exist in the primitive streak remains unclear.

In this study, we use genetic tracing of *T-* and *Foxa2*-expressing cells and find there is an orderly allocation of primitive streak cells first into the left ventricle progenitors at mid-streak stage, then the right ventricle progenitors at late streak stage, and finally at the no bud (OB)–early bud (EB) stage into outflow tract and atrial progenitors. Consistent with this, we identified independent sets of *Foxa2+* cells allocated to the left ventricle and right ventricle. Allocation of cells to the outflow tract and atria happens at similar gastrulation stages but from distinct locations within the primitive streak. The outflow tract originates from distal regions of the primitive streak, while proximal regions contribute to the atria. Moreover, the outflow tract forms from primitive streak cells that initially expressed *Foxa2* but subsequently turned off *Foxa2* expression as they switched their contribution from the right ventricle to the outflow

tract. Crucially, by combining single-cell transcriptomic assays with a lineage tracer to label cells supplying only the poles of the heart, we uncover molecularly distinct subpopulations of cells that correspond to progenitors for right ventricle, outflow tract, and atria in the SHF. Further single-cell transcriptomic experiments established that the primitive streak cells contributing to the ventricles and outflow tract/atria are also molecularly distinct. Thus, rather than a simple subdivision of cardiac progenitors into FHF and SHF, our analysis reveals a more elaborate map for the source of cells that form the heart with distinct spatial and temporal origins for outflow tract and atrial progenitors as well as left and right ventricular progenitors. We conclude that the cardiac progenitors are prepatterned within the primitive streak, and this prefigures their contribution to distinct anatomical structures of the heart both in time and space. These results have implications for the classification of congenital heart diseases based on the origin of malformation in a specific mesodermal lineage and for the design of in vitro methods to generate specific cardiac cells from pluripotent stem cells.

## Results

### Genetic tracing of primitive streak cells using a tamoxifen-inducible T reporter

During gastrulation, epiblast cells ingress through the primitive streak to form the mesoderm. This process occurs over an extended period, from embryonic day (E)6 to E8 in the mouse. The T-box transcription factor $T$ is expressed in the primitive streak and is down-regulated shortly after ingression during migration within the nascent mesoderm. To assess the developmental time points at which the $T$-expressing primitive streak cells are destined to contribute to the heart, we performed genetic tracing using an inducible $T^{nGFP-CreERT2/+}$ mouse, expressing $CreERT2$ and nuclear localised GFP (nGFP) downstream of the endogenous $T$ [16], in combination with the $R26R^{tdTomato/+}$ reporter mice ($T^{nEGP-CreERT2/+}$; $R26R^{tdTomato/+}$, Fig 1A) [17].

We first assessed how long tamoxifen activity persists after administration by oral gavage in a pregnant mouse. We administered tamoxifen (0.08 mg/bw) at E5, approximately 24 hours before the initial onset of $T$ expression in the primitive streak [18]. We detected recombined tdTomato-expressing cells within mesodermal derivatives including the heart tube, head mesenchyme, and endothelium, albeit at a low density, in E8.5 embryos (S1A Fig). Thus, the activity of the tamoxifen persists for at least 24 hours at this dose.

### Primitive streak cells contribute first to the ventricles and subsequently to the outflow tract and atria myocardium

We next administered single doses of tamoxifen $T^{nEGP-CreERT2/+}$; $R26R^{tdTomato/+}$ at successive stages of gastrulation. We reasoned this would result in pulse-labelling of most populations of $T$-expressing cells from the time of administration for at least 24 hours (Fig 1B). The fate of their progeny could then be followed and the developmental stages during which primitive streak cells contribute to the left and right ventricle, atria, and outflow tract could be deduced. To gain better control over the embryonic stages, we synchronised mice in estrus and mated them over short periods of 2 hours for all the experiments (from 7 AM to 9 AM; vaginal plugs were checked at 9 AM and positive were defined as E0). To quantify the contribution of the $T$-expressing cells to the heart, we measured the surface area occupied by the tdTomato-positive cells in the heart myocardium at E12.5 and within each cardiac chamber.

An early administration of tamoxifen at E6+8h resulted in the population of left and right ventricles, outflow tract, and atria with tdTomato-expressing cardiomyocytes (in 6 out of 6

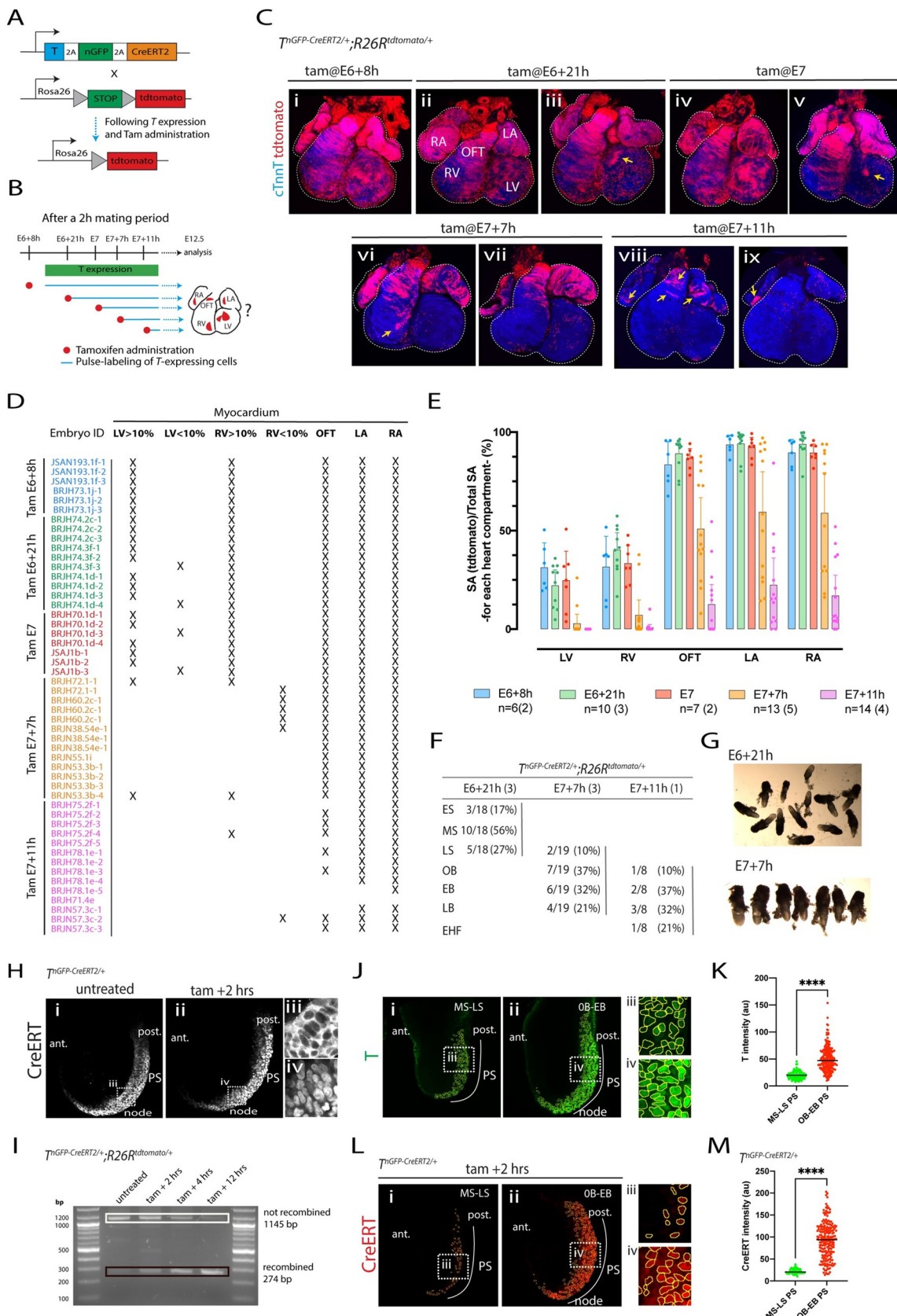

**Fig 1. Genetic tracing of the *T+* primitive steak cells. (A)** Schematics of the $T^{nGPF-CreERT2/+}$; $R26R^{tdTomato/+}$ allele [16]. Cre-ERT2 is expressed in cells expressing *T*. In the presence of tamoxifen, Cre protein is translocated to the nucleus where it recombines the $R26R^{tdTomato/+}$ reporter. As a result, the cell and its descendants are permanently labelled. **(B)** Diagram of the experimental approach. *T*-expressing cells and their descendants are labelled, from E6+21h, E7, E7+7h and E7+11h by administrating a dose of tamoxifen (Tam) to $T^{nGPF-CreERT2/+}$; $R26R^{tdTomato/+}$ mice (0.08 mg/body weight via oral gavage). Cell descendants in the myocardium are analysed at E12.5. **(C)** Representative hearts resulting from the administration of tamoxifen at different time points in $T^{nGPF-CreERT2/+}$; $R26R^{tdTomato/+}$ immunostained with cTnnT to reveal the cardiomyocytes (blue). Yellow arrows identify small patches of tdTomato positive cardiomyocytes in the LV (iii and v), in the RV (v), and in the outflow tract and atria (vii and viii). Views are ventral. Single epicardial cells are labelled in each condition. **(D–E)** Summary of all $T^{nGPF-CreERT2/+}$; $R26R^{tdTomato/+}$ hearts examined. The contribution of the *T*-expressing cells to the different compartments of the heart is quantified by measuring the proportion of tdTomato-positive myocardium. Numbers in brackets in (E) represent the number of litters assessed. Error bars are SD. The data underlying (D–E) can be found in S1 Source Data. **(F)** Stage variation quantified according to Downs and Davies criteria [20]. Number in brackets represents the number of litters assessed. All the timed matings were for 2-hour periods. **(G)** Embryos collected at E6+21h or E7+7h showing variation in stage. **((H)** Representative $T^{nGPF-CreERT2/+}$ embryos untreated or tamoxifen treated for 2 hours mice (0.08 mg/body weight via oral gavage) and immunostained with oestrogen receptor. Insets (iii) and (iv) are magnified view from (i) and (ii), respectively. **(I)** PCR amplicons generated from the genomic region in which Cre-mediated recombination occurs, resolved on an agarose gel. Before recombination, the PCR product is 1,145 bp (white rectangle); after recombination, it is 274 bp (black rectangle). Template gDNA was extracted from either an ear clip of an adult $T^{nGPF-CreERT2/+}$; $R26R^{tdTomato/tdTomato}$ mouse (untreated) or $T^{nGPF-CreERT2/+}$; $R26R^{tdTomato/tdTomato}$ embryos dissected at 2, 4, and 12 hours following oral gavage with Tamoxifen, as labelled. An increase in the proportion of the recombined band can be seen over time following Tamoxifen administration. The data underlying (I) can be found in S1 Raw image. **(J)** Representative $T^{nGPF-CreERT2/+}$ embryos at the MS-LS and OB-EB stages, immunostained with. (iii) and (iv) are magnified views in (i) and (ii), respectively. **(K)** Quantification of T intensity in single segmented nuclei from embryo shown in (J). The data underlying (K) can be found in S2 Source Data. **(L)** Representative $T^{nGPF-CreERT2/+}$ embryos at the MS-LS and OB-EB stages treated with tamoxifen for 2 hours (0.08 mg/body weight via oral gavage) and immunostained for oestrogen receptor. (iii) and (iv) are magnified views in (i) and (ii), respectively. **(M)** Quantification of Oestrogen receptor intensity in single segmented nuclei from the embryos shown in (L). The data underlying (M) can be found in S2 Source Data. Embryos in J (i) and (ii) and in L (i) and (ii) were immunostained together and imaged under similar conditions. ant., anterior; cTnnT, cardiac troponin T; EB, "early bud" stage; EHF, early head fold; LA, left atria; LB, "late bud stage"; LS, late-streak; LV, left ventricle; MS, mid-streak; OB, "no bud" stage; OFT, outflow tract; post., posterior; PS, primitive streak; RA, right atria; RV, right ventricle. Scale bar: 200 μm.

hearts analysed; Fig 1Ci, 1D, and 1E). The contribution of tdTomato-positive cardiomyocytes to the left and right ventricle was similar (left ventricle: 31.4% ± 12.4; right ventricle: 31.6% ± 15.5, mean ± SD, *p*-value: ns). Contribution to the outflow tract and atria was the highest, covering almost their entirety in all cases (83.8% ± 11.7 and 91.7% ± 5.9, mean ± SD, for outflow tract and atria, respectively).

An administration at E6+21h or E7 also resulted in tdTomato-expressing cardiomyocytes in the left and right ventricles, outflow tract, and atria (in 17 out of 17 hearts analysed; Fig 1Cii–1Cv, 1D, and 1E). In these cases, the contribution of the tdTomato-positive cells to the left ventricle was lower, on average, than to the right ventricle (for E6+21h, left ventricle: 22.2% ± 11.1; right ventricle: 40.2% ± 12.2 and for E7, left ventricle: 24.8% ± 16, and right ventricle: 33.4% ± 9.9, mean ± SD, *p*-value: 0.004). Contribution to the outflow tract and atria continued to be the highest in all cases (89.1% ± 8.4 and % 94.1 ± 6.9, mean ± SD, for outflow tract and atria, respectively). Variability in embryonic stages within litters at the time of tamoxifen administration can explain the variability in the results (see below). For example, in 4/17 hearts, tdTomato-expressing cardiomyocytes populated less than 10% of the total left ventricle surface area. In those hearts, the proportion of tdTomato-positive cardiomyocytes found in the right ventricle was high (Fig 1Ciii, 1Cv, and 1D). We never observed the reverse where a lower proportion of tdTomato-positive cardiomyocytes was found in the right ventricle compared to the left ventricle (Fig 1D). These results suggest that subsets of the embryos at E6+21h and E7 were at more advanced embryonic stages, with most left ventricle precursors having already left the primitive streak with right ventricular precursors remaining in the primitive streak.

Next, we asked if tamoxifen administration at a later time point would result in the absence of tdTomato-expressing cardiomyocytes in the left ventricle and right ventricle. We administered tamoxifen at E7+7h. tdTomato-expressing cardiomyocytes were detected in the left

ventricle in only 2/13 hearts (Fig 1D). tdTomato-positive cardiomyocytes were detected more frequently in the right ventricle, in 7/13 hearts (Fig 1D). However, these cells covered less than 5% of the total right ventricle surface area (yellow arrow in Fig 1Cvi and 1E), except for the 2 hearts in which we also detected tdTomato-expressing cardiomyocytes in the left ventricle. These results indicate that left ventricle and most right ventricle precursors have left the primitive streak by E7+7h. In these embryos, tdTomato-expressing cardiomyocytes were evident in the outflow tract and atria in 13/13 hearts, covering in some cases between 72.8% and 94% of the total surface area of the outflow tract and atria, including hearts without tdTomato-expressing cardiomyocytes in the left ventricle and right ventricle. These results suggest that most outflow tract and atria precursors are still located in the primitive streak at stages after the ventricular precursors have already left the primitive streak.

Finally, when tamoxifen was administered at a later stage (E7+11h), the contribution of the tdTomato-expressing cardiomyocytes to the outflow tract and atria was lower (Fig 1Cviii–1Cix and 1E, 12.6% ± 17.4 and 19.7% ± 20.8). In all cases, these embryos lacked tdTomato-expressing cells in the left and right ventricles. In 5/14 hearts, tdTomato-expressing cells were absent in the outflow tract but present in the atria, albeit in low numbers. The reverse, i.e., tdTomato-expressing cells found only in the outflow tract but not in the atria, was never observed, suggesting that the atrial precursors are the last cardiomyocyte precursors to leave the primitive streak.

Combining the $T^{nEGP-CreERT2/+}$ transgene with the R26$^{mGFP/+}$ reporter [19] produced similar results (S2A Fig). An early tamoxifen administration (at E6+8h) led to mGFP-positive cardiomyocytes populating the left ventricle myocardium. When we administered tamoxifen late (E7+7h), no mGFP cardiomyocytes were identified in the left ventricle myocardium. We observed mGFP cardiomyocytes in the outflow tract and atria in both conditions.

To analyse variability in embryonic stages at the time of tamoxifen administration, we dissected litters of $T^{nEGP-CreERT2/+}$; $R26R^{tdTomato/+}$ embryos at E6+21h and E7+7h and staged embryos according to morphological landmarks using the dissecting brightfield microscope [20]. As expected, we found a range of stages for each time point. Litters dissected at E6+21h included early (17%), mid (56%), and late-streak stages (27%). Litters dissected at E7+7h included late streak (10%), OB (37%), EB (32%), and late bud stages (LB, 21%) (Fig 1F and 1G). Whole-mount immunostaining for the oestrogen receptor ERT revealed CreErt2 in the nucleus within 2 hours of oral gavage of tamoxifen (0.08 mg/bw) (Fig 1H and S3Ai–S3Avi Fig). PCR analysis showed that the $R26R^{tdTomato/+}$ locus was recombined (Fig 1I and S4Ai–S4Aii and S4B Fig).

Together, these results indicate that cardiac progenitors ingress in an orderly sequence through the primitive streak. Left ventricle progenitors leave the primitive streak first, at the mid-streak stage, followed by right ventricle progenitors at the late-streak stage. Outflow tract and atrial precursors leave the primitive streak at subsequent stages, starting around the OB stage.

## *T* is expressed at a low level in ventricular progenitors and at higher levels in atrial and outflow tract progenitors

Ventricular progenitors were captured less efficiently than the outflow tract and atria progenitors in *T*-lineage tracing experiments. This prompted us to quantify T and Cre protein expression levels in embryos collected at the mid to late streak stages, when ventricular progenitors reside in the streak, and OB to EB stages, when outflow tract and atria progenitors reside instead in the streak. All primitive streak cells express *T*. However, we found that primitive streak cells at the mid to late streak stages express lower levels of T protein compared to the

OB and EB stages (Fig 1Ji–1Jiv and 1K). The transgenic $T^{nEGP\text{-}CreERT2/+}$ embryo showed a similar pattern, with lowest Cre (Fig 1L and 1M) and nGFP signal at the mid-late streak stage (compare GFP levels in S4Ai and S4Aii Fig). These results suggest why not all ventricular progenitors are captured by our $T$-lineage tracing experiments.

### Late primitive streak cells/atrial progenitors contribute to posterior regions of the first heart field and differentiate early into cardiomyocytes

The FHF contributes mainly to the left ventricle in addition to a minority of the atria [21,22], and our late $T$-lineage tracing at E7+7h may have missed these FHF atrial progenitors. This prompted us to test whether the descendants from the late E7+7h also populated the FHF. We assayed cardiac troponinin T (cTnnT), to mark FHF cells, since these differentiate early into cardiomyocytes to establish the cardiac crescent. We found in 5 out of 9 embryos analysed, sparse tdTomato-expressing cardiomyocytes located in posterior regions of the cardiac crescent or prospective inflow (Fig 2Ai–2Aiv). No tdTomato-expressing cardiomyocytes were found in anterior regions of the cardiac crescent in any of the embryos examined. tdTomato-positive cells could also be found in the endoderm (blue arrows in Fig 2Aii). Furthermore, analysis at the heart tube stage reveals that late E7+7h $T$ cell descendants were populating the inflow regions of the heart tube (in 3 out of 4 embryos; Fig 2Bi and yellow arrows in Fig 2Bii). A sparse contribution to the endocardium was also noted (blue arrows in Fig 2Biii). We conclude that a subset of the late $T$-expressing cells/atrial progenitors are recruited to posterior regions of the FHF and differentiate early into cardiomyocytes.

### Independent *Foxa2*-expressing cells located in the anterior primitive streak at the mid to late streak stages contribute to either the left or right ventricular myocardium

Genetic tracing experiments at E6-E6.5 [8,23] showed that *Foxa2*-positive cells contribute to cardiomyocytes in the ventricles but not in the atria. These results prompted us to test whether *Foxa2*-positive cells contribute to ventricular myocardium at similar stages to the $T$-expressing cells, i.e., at the mid-late streak stage. We performed genetic tracing of the *Foxa2*-positive cells using a $Foxa2^{nEGP\text{-}CreERT2/+}$ line expressing *CreERT2* downstream of the endogenous *Foxa2* combined with the $R26R^{tdTomato/+}$ reporter line [16] (Fig 3A and 3B). Mice were synchronised in estrus, and timed matings performed for 2 hours to better control embryonic stages.

Administration of tamoxifen at E6+21h resulted in patches of tdTomato-expressing cardiomyocytes populating the left and right ventricle and outflow tract (Fig 3Ci). On average, the contribution of the tdTomato-positive cells to the ventricles was low (left ventricle: 9.5% ± 9.5; right ventricle: 11.2% ± 10, $p$-value: ns). In 5 hearts out of 14, less than 10% of the left or right ventricle myocardium was populated by tdTomato-expressing cardiomyocytes. In addition, in 3 out of 14 hearts, no tdTomato-expressing cells could be detected in either the left or right ventricles (Fig 3D and 3E).

When we administered tamoxifen at E7+7h, tdTomato-positive cardiomyocytes could be detected in the left ventricle in only 1 out of 11 embryos and in the right ventricle in 3 out 11 embryos (Fig 3Cii, 3D, and 3E). These cells covered less than 10% of the total left and right ventricle surface area (Fig 3D and 3E). In 3/14 hearts, small tdTomato-expressing domains formed either in the left (when labelled at E6+21h, $n = 2$) or right ventricular myocardium (when labelled at E7+7h, $n = 1$). This indicates that independent groups of *Foxa2*-expressing progenitors exist in the primitive streak that contribute to the left and right myocardium (Fig 3Civ and 3Cv and 3E). We conclude that *Foxa2*-positive primitive streak cells contribute to

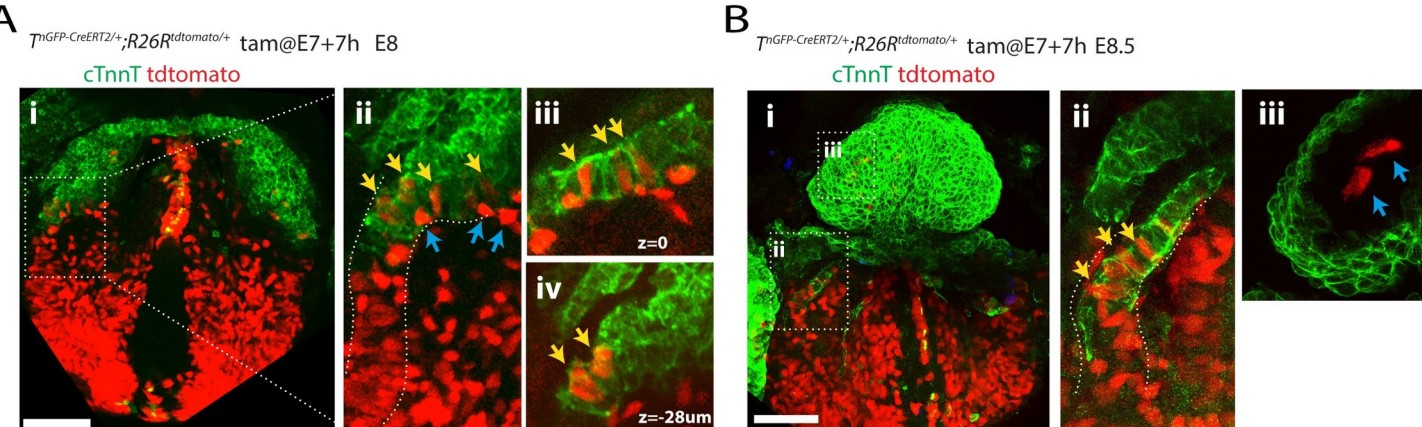

**Fig 2. Descendants from late E7+7h T-positive primitive streak cells contribute to posterior regions of the FHF. (A, B)** Representative embryos resulting from the administration of tamoxifen at E7+7h in $T^{nGPF-CreERT2/+}$; $R26R^{tdTomato/+}$ immunostained with cTnnT to reveal the cardiomyocytes of the first heart field (green) (A) and heart tube (B). Yellow arrows identify tdTomato-positive cardiomyocytes in the posterior first heart field (Aii–Aiv) and inflow tract (Bii). Blue arrows identify tdTomato-positive endodermal cells (Aii) and endocardial cells (Biii). (Aii) shows magnified view of inset in (Ai). (Ai–Aii and Bi–Bii) are z-maximum projection, (Aiii–Aiv and Biii) are single optical sections. Views are ventral. Single epicardial cells are labelled in each condition. Scale bar: 200 μm.

the ventricular myocardium at similar stages but in a lower proportion to the *T*-expressing cells.

Analysis of the embryos at stages when the cardiac crescent is differentiating into cardiomyocytes confirmed that cells arising from *Foxa2*-positive cells at E6+21h contribute to the myocardium and other lineages including the epicardium and pericardium (S5A Fig). This is consistent with previous studies showing that the FHF contribute to all these cell types [6,8,24,25]. We found no contribution to the endocardium located below the presumptive myocardial epithelium, however (S5B Fig).

These results prompted us to analyse the location of *Foxa2*- and *T*-expressing cells at the mid and late streak stages. Whole-mount immunostaining for T and Foxa2 revealed coexpression in individual primitive streak cells at the mid-streak position of mid and late streak stages. Double positive cells are also detected in the definitive endoderm (Fig 4Bi and 4Bii and 4Ci and S6Ai and S6ii Fig) and at the distal tip of the embryo where the node forms (Fig 4Cii). These results indicate that *T* and *Foxa2* are not always in mutually exclusive cell populations as previously reported [26]. Instead, it suggests *Foxa2* and *T* are coexpressed in a population of aPS cells that contribute to the ventricular myocardium but not to the atria. Finally, Foxa2 is undetectable in the primitive streak from the OB stage onwards, the stages during which primitive streak cells switched their contribution from the ventricles to the outflow tract and atria (Fig 4E and 4F and S6D–S6"i Fig).

## All ventricular progenitors derived from the *Foxa2* lineage express *Foxa2* at the mid-late streak stage

It is possible that a subset of the ventricular progenitors expresses *Foxa2* early and down-regulate *Foxa2* and up-regulate *T* by the mid to late streak stages. To test this idea, we administered tamoxifen in *Foxa2*$^{nEGP-CreERT2/+}$ $R26R^{tdTomato/+}$ embryos earlier (E6+8h), collected the embryos at E6+21h, and assayed for Foxa2 and T protein. Because of the variation in embryonic stages, we could analyse both the early and mid to late streak stages (Fig 1F and 1G). As expected, at the early streak stage, we found Foxa2-positive cells with weak tdTomato signal located in the distal primitive streak (yellow arrows in Fig 5A). At the mid-streak stage, the

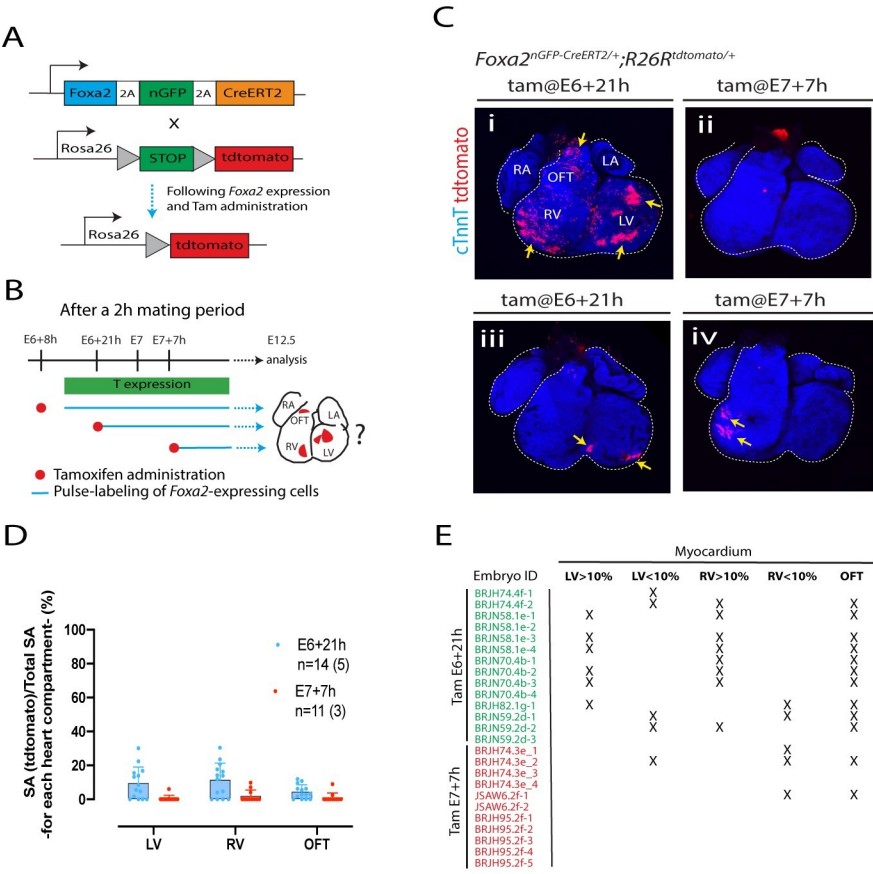

Fig 3. Independent *Foxa2+* progenitors contribute to the ventricles and outflow tract. **(A)** Schematics of the *Foxa2^{nGPF-CreERT2/+}; R26R^{tdTomato/+}* allele [16]. **(B)** Pulse-labelling of the *Foxa2*-expressing cells and their descendants, from stage (i) E6+21h, (ii) E7+7h by administration of tamoxifen to *Foxa2^{nGPF-CreERT2/+}; R26R^{tdTomato/+}* mice (0.08 mg/ body weight via oral gavage). Cell descendants in the heart analysed at E12.5. **(C)** Representative hearts resulting from the administration of tamoxifen at different time points in *Foxa2^{nGPF-CreERT2/+}; R26R^{tdTomato/+}* immunostained for cTnnT to reveal the cardiomyocytes (blue). Views are ventral. Single epicardial cells are also labelled in (iii). **(D, E)** Summary of all hearts analysed. The contribution of the *Foxa2*-expressing cells to the different compartments of the heart is quantified by measuring the proportion of tdTomato-positive myocardium. Numbers in brackets in (E) represent the number of litters assessed. Error bars are SD. Scale bar: 200 μm. The data underlying (D, E) can be found in S3 Source Data. cTnnT, cardiac troponin T; LA, left atria; LV, left ventricle; OFT, outflow tract; RA, right atria; RV, right ventricle.

streak had extended along its proximal-distal axis, and we found tdTomato-positive cells coexpressing Foxa2 and T in the aPS (Fig 5B–5G). We also found rare tdTomato-positive cells expressing only T and not Foxa2 (red arrow in Fig 4Cii, yellow arrows in Fig 3F). Quantification of Foxa2 and T intensities revealed that the tdTomato positive cells located in the aPS coexpress low levels of Foxa2 and T (yellow arrows in Fig 5Ci–5Ciii, Fig 5D–5F and S7A Fig). The tdTomato-positive cells located most distally expressed a higher level of Foxa2 and low or no T (blue arrows in Fig 5Cii and 5Ciii and Fig 5D–5F). These more distal regions correspond to where the axial mesoderm and definitive endoderm are established [27,28]. We conclude that cells expressing *Foxa2* early are not contributing to proximal regions of the primitive streak. Instead, these cells maintain *Foxa2* expression in the aPS and contribute to the left and right ventricles at the mid-to-late streak stages (Fig 5G and 5H).

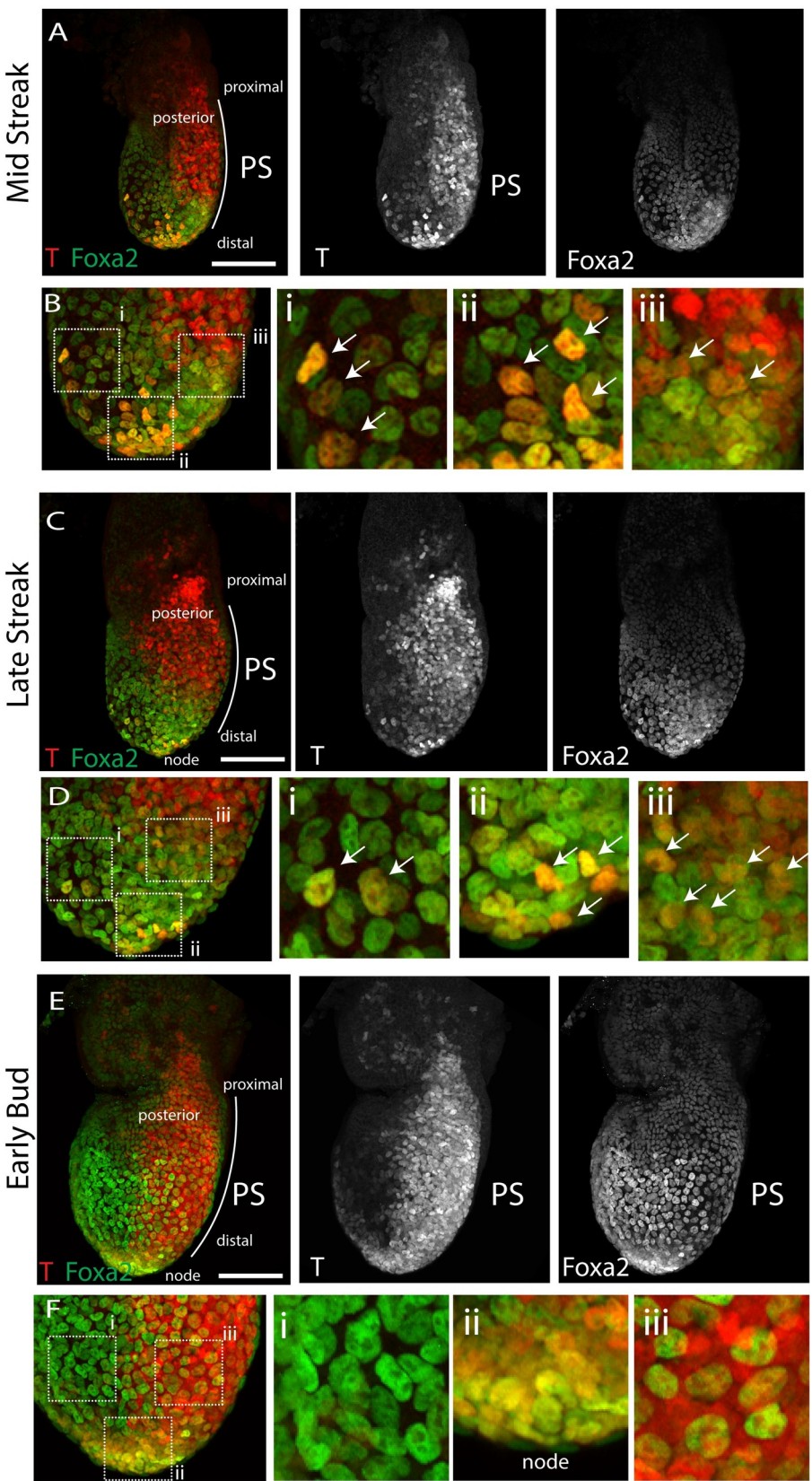

**Fig 4. T and Foxa2 colocalise in primitive streak cells.** Representative embryos of E6+21h MS (A, B) and LS (C, D) and E7+7h EB (E, F). Embryos are immunostained for T (red) and Foxa2 (green). Images are z-maximum projections. Views are lateral/slightly posterior to show the whole width of the primitive streak. Insets in Bi–Biii, Di–Diii, and Fi–Fiii show magnified views and white arrows point to T+/Foxa2+ double positive cells. Scale bar: 100 μm. EB, "early bud" stage; LS, late-streak; MS, mid-streak; PS, primitive streak.

## Distal primitive streak cells down-regulate *Foxa2* expression as they switch their contribution from the ventricles towards the outflow tract myocardium

Following tamoxifen administration in *Foxa2*$^{nEGP-CreERT2/+}$ *R26R*$^{tdTomato/+}$ embryos at E6+21h instead, we found tdTomato-expressing cells located in the outflow tract myocardium (Fig 3Cii and 3Di and S5A Fig). As previously mentioned, when we administered tamoxifen at E7+7h, no tdTomato-positive cardiomyocytes could be detected in the outflow tract (in 11 out of 13 hearts; Fig 3Ciii). This could indicate that a subset of the *Foxa2*-positive primitive streak cells, at E6+21h, are outflow tract progenitors and not solely ventricular progenitors. Alternatively, primitive streak cells contributing to the outflow tract may arise from cells that had previously expressed *Foxa2* but no longer do so as the primitive streak switches its contribution from right ventricular to outflow tract myocardium by E7+7h. Consistent with this latter hypothesis, the presence of tdTomato-expressing cardiomyocytes in the outflow tract was always associated with tdTomato-expressing cardiomyocytes in the right ventricle (in 11/14 hearts; Fig 3E).

We next administered tamoxifen in *Foxa2*$^{nEGP-CreERT2/+}$ *R26R*$^{tdTomato/+}$ embryos at E6+21h and fixed the embryos at E7+7h (Fig 6A). We immunostained embryos for Foxa2 and T protein. Computationally masking cells for Foxa2 expression (i.e., endodermal and axial mesoderm cells; Fig 6Aii–6Aiii) revealed tdTomato-expressing mesodermal cells (i.e., cells that had expressed *Foxa2*) located in distal regions of the primitive streak and the medial/dorsal mesoderm (Fig 6Aiii). No tdTomato-positive cells were detected in lateral regions of the mesoderm.

We then analysed all tdTomato-expressing cells within the distal primitive streak region. We found these cells expressed T protein (yellow arrows in Fig 6Av), although none coexpressed Foxa2 and T. Cells expressing Foxa2 but not T corresponded to endodermal cells, as indicated by their location in the outer layer of the embryo (green arrow in Fig 6Av). These experiments show that the outflow tract myocardium descendants from *Foxa2* progeny arise from the distal primitive streak. The outflow tract progenitors are *Foxa2* negative, but they arise from cells that expressed *Foxa2* previously. We propose that distal primitive streak cells down-regulate *Foxa2* as they switch their contribution from the right ventricle to the outflow tract myocardium (Fig 6C). Atrial cells arise instead from the proximal primitive streak since they never arise from cells that had expressed *Foxa2*, and all the cells located in the aPS express *Foxa2* (Fig 5E and 5F).

We further characterised the tdTomato-expressing mesodermal cells arising from the *Foxa2* lineage. These cells have a cranial mesodermal identity since they express *Mesp1* (S8A Fig). In addition, they are located adjacent to endoderm expressing the nodal and Wnt antagonists Cerebus1 (Cer1) and Dkk1, respectively (S8B and S8D Fig) [29]. The primitive streak and posterior mesoderm are instead marked by Wnt/β-catenin signalling activity (S8E Fig). tdTomato-positive cells located in anterior regions of the embryo (red arrows in S8C Fig) were unlikely to be endothelial cells since they did not express the endothelial marker Flk1 (for all the embryos we analysed (*n* = 7)). Instead, they are more likely to be cardiac progenitors since they were responding to BMP signalling (by E7+12h), and cardiomyocytes of the FHF—or cardiac crescent—are characterised by high levels of BMP signalling activity at these stages [11,30] (Fig 6B and S9A–S9A" Fig).

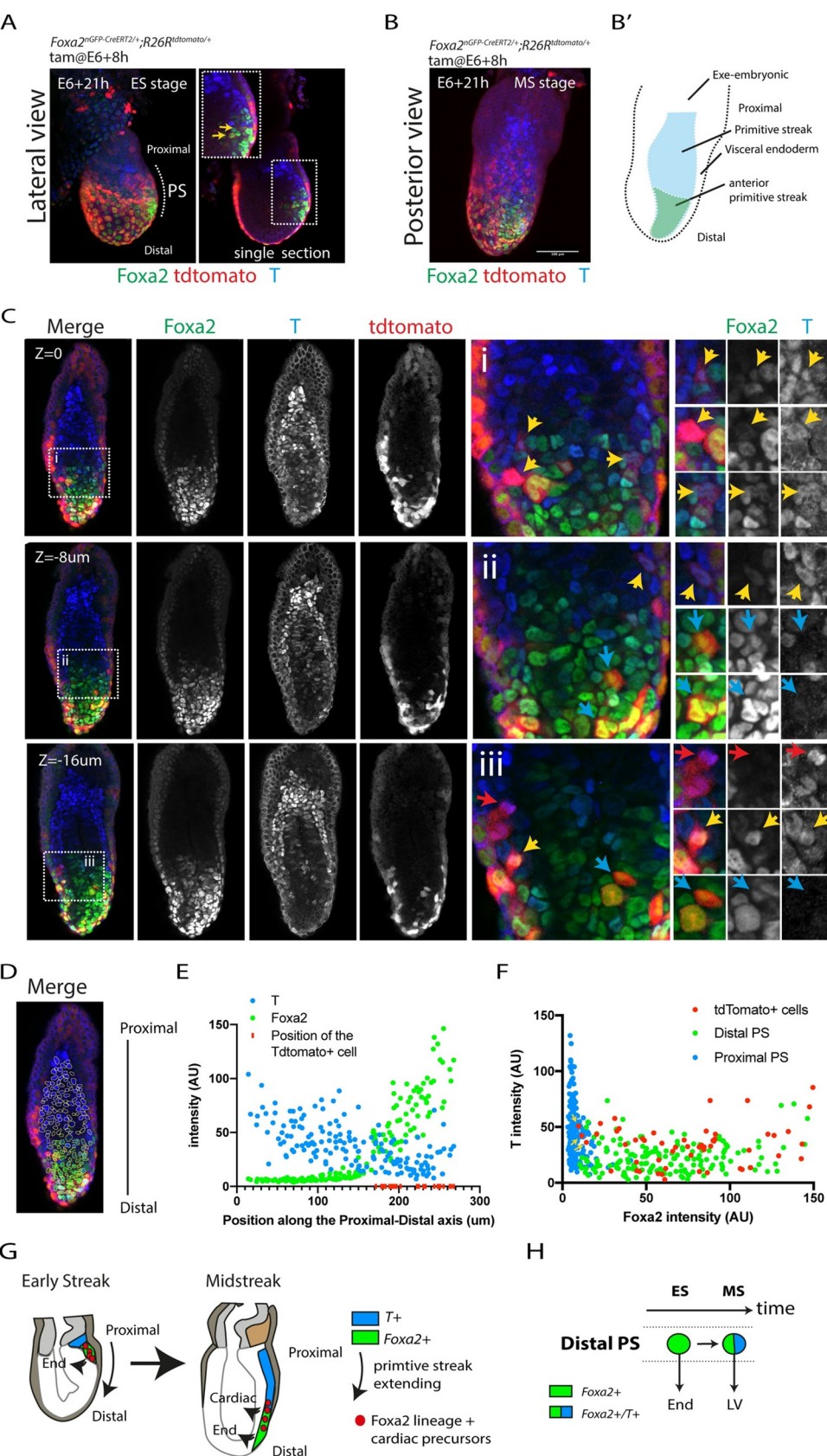

**Fig 5. *Foxa2* lineage-positive cells maintain *Foxa2* expression at the mid-late steak stage. (A, B)** Representative embryos at the early (A) and mid-late (B) streak stages resulting from the administration of tamoxifen at E6+8h in *Foxa2^nGPF-CreERT2/+; R26R^tdTomato/+* immunostained for *Foxa2* (green) and T (blue). View is lateral in (A) and posterior in (B). (B") Schematics of the embryo shown in (B). (**C**) Same embryo as in (B). Single optic sections are shown. (i–iii) are magnified views of the insets. Yellow arrows point to double Foxa2/T positive, blue arrows to Foxa2 positive T negative, and red arrows to Foxa2 negative T positive tdTomato-expressing cells. (**D**) Representative example of a segmented image based on Foxa2 and T signal. (**E**) Quantification of T and Foxa2 signal intensities along the proximal-distal axis of the embryo in single cell nuclei. All the tdTomato-positive cells are all located in the distal (anterior) portion of the primitive streak. The data underlying (E) can be found in S4 Source Data. (**F**) Quantification of Foxa2 and T intensities in single cell in distal (green), proximal (blue) primitive streak cells and tdTomato-positive cells. The data underlying (F) can be found in S4 Source Data. (**G, H**) Model. As the primitive streak extend along the proximal-distal axis, from the early to mid-late streak stages, a subset of the *Foxa2*-positive cells switch contribution from endoderm to cardiac (ventricles) progenitors and express *T*. ES, early streak; LV, left ventricle; MS, mid-streak; PS, primitive streak.

## Ventricular, outflow tract, and atrial progenitors have distinct locations and transcriptional profile within the heart fields

Having identified distinct populations of *T*-expressing cells descendants, we reasoned their final locations could correlate with the previously described heart fields at gastrulation stages [11]. We analysed the locations of the descendants of early E6+21h and late E7+7h *T*-expressing cells. These cells formed the ventricles and most of the outflow tract/atria, respectively (see above). We used a BMP reporter (S9A–S9A" Fig) and assayed for Phospho(P)-Smad1/5/8 to mark the FHF and for Raldh2 to mark the posterior SHF (Fig 7D). Progenitors derived from the early E6+21h *T*-expressing cells contributed to anterior structures, distinct from the region occupied by late E7+7h *T*-expressing cells (Fig 7Ai and 7Aii). The descendants from the late E7+7h were located in posterior mesoderm anterior to the node where Raldh2 localisation is strong (Fig 7Di–7Div). They were instead mostly excluded from the FHF domain in which BMP activity is high and also from the cranial paraxial mesoderm with no detectable BMP activity (Fig 7Biii–7Biv and 7Ci and 7Cii and S10A Fig). Some tdTomato-expressing cells were also sparsely intermingled within the posterior BMP-positive domain. Among these, some had detectable BMP activity (see yellow and red arrows in Fig 7Bi–7Biv and 7Ci and 7Cii and S10A Fig). This is consistent with late recruitment of E7+7h progenitors to posterior regions of the cardiac crescent (see above).

To ask whether molecular differences exist between the ventricular, outflow, and atrial cardiac progenitors, we performed single-cell transcriptome analysis of 4 *T^nEGP-CreERT2/+ R26R^tdTomato/tdTomato* embryos dissected 7 hours after tamoxifen administration at E7+7h (early head fold (EHF) stage; Fig 8A). We reasoned we could use the presence or absence of tdTomato transcripts to discriminate the transcriptome of the tdTomato-positive outflow tract and atrial progenitors from the tdTomato-negative ventricular progenitors within the heart fields. We obtained 3,494 high-quality single-cell transcriptomes. We clustered cells into subpopulations, guided by available single-cell transcriptomic data ([3]; see Methods, Fig 8Bi, S11A–S11D and S12A–S12D Figs). Within these subpopulations, cardiac progenitors belonging to the previously defined FHF, anterior heart field (AHF), and posterior second heart field (pSHF) [11] and cranial paraxial mesoderm [31–35] were identified based on the expression of *Tbx5*, *Tbx20*, and *Hand2* for FHF; *Tbx1* and *Fgf8/10* for AHF; *Raldh2*, *Hoxb1*, and *Tbx6* for pSHF; and *Pitx2*, *Alx1*, and *Cyp26a1* for the cranial paraxial mesoderm (Fig 8B and 8Cii). Notably, *Fgf8* expression was reduced in the cranial paraxial mesoderm cluster, consistent with the absence of *Fgf8* expression in this cell population [36].

We next analysed the proportion of tdTomato-positive cells in each cluster (S12C Fig, Fig 8Di–8Diii and S6 Source Data). Consistent with the imaging, we found that posterior mesoderm populations (i.e., caudal, presomitic, intermediate, and paraxial mesoderm) and

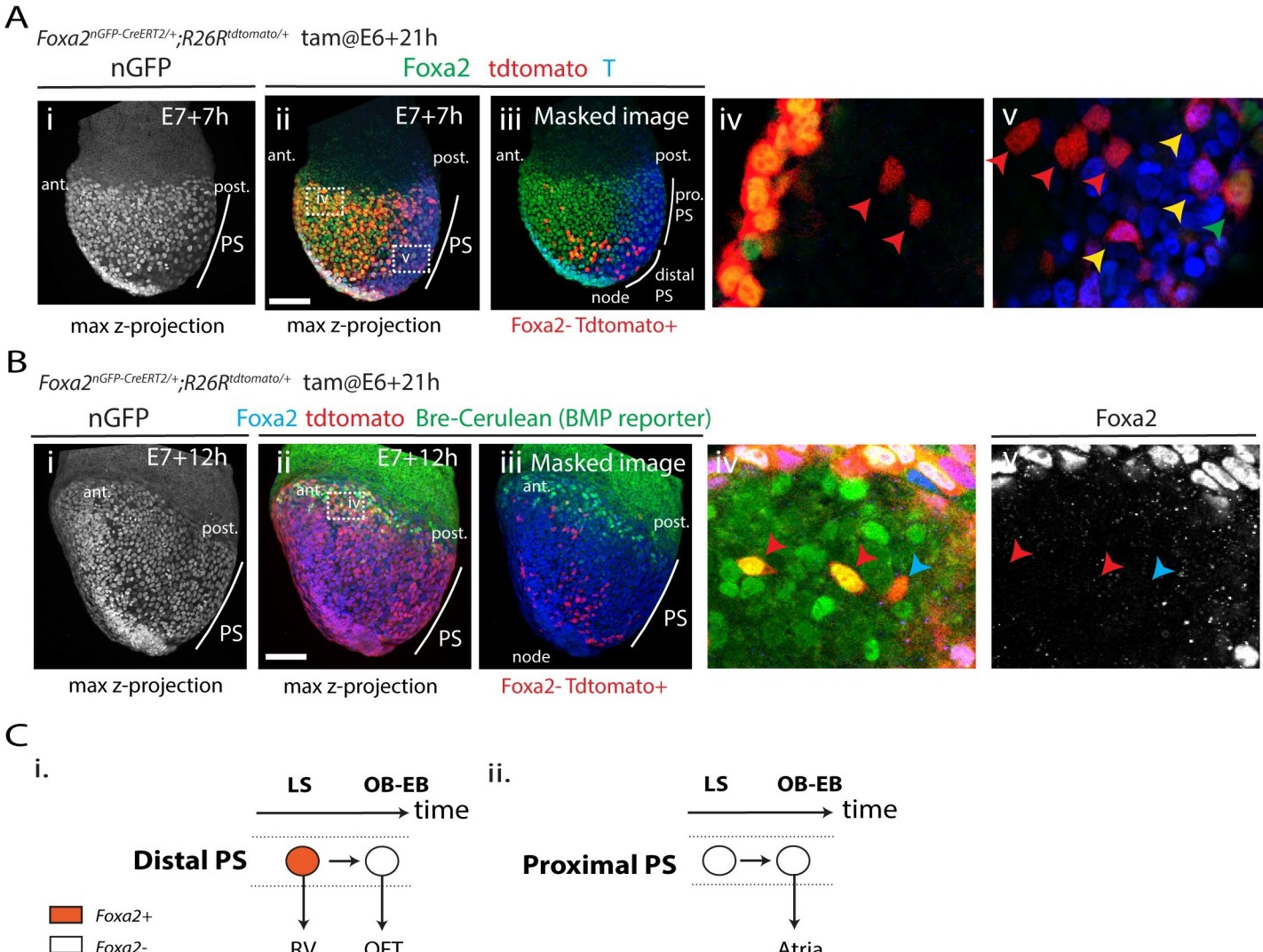

**Fig 6. Ventricular precursors originate from distal regions of the primitive streak. (A)** Localisation of the tdTomato+ cells from the *Foxa2* lineage are assessed (i, ii) at E7+7h (OB-EB stage). nGFP signal from the transgene is shown in (i). The embryos are immunostained for T (blue) and Foxa2 (green) (ii). Image is a z-maximum projection. (iii) The same embryo (i, ii), but with Foxa2+ cells masked to reveal the tdTomato+/Foxa2− cells. Insets (iv) and (v) show magnified views in single optical sections of (ii). Red arrows point to tdTomato+/Foxa2−/T− mesodermal cells. Yellow arrows point to tdFtomato+/Foxa2−/T+ primitive streak cells. Green arrows point to tdTomato+/Foxa2+/T− endodermal cells. No tdTomato+/Foxa2+/T+ cells are identifiable. **(B)** Localisation of the tdTomato+ cells from the *Foxa2* lineage are assessed in Bre-cerulean (BMP reporter) embryos at (i, ii) E7+12h (EHF stage). nGFP signal from the transgene is shown in (i). The embryos are immunostained for Foxa2 (blue) (ii). Image is a z-maximum projection. (iii) The same embryo (i, ii), but with Foxa2+ cells masked to reveal the tdTomato+/Foxa2− cells. Insets (iv) and (v) show magnified views in single optical sections of (ii). Red arrows point to tdTomato+/Foxa2−/Bre-cerulean+ mesodermal cells. Blue arrow points to tdTomato+/Foxa2−/Bre-cerulean− cell. **(C)** (i) A common progenitor between the RV and the outflow tract in the distal primitive streak. *Foxa2* is down-regulated as it switches its contribution towards outflow tract myocardium. (ii) Proximal PS contribute to the atria and are descended from cells that had not expressed *Foxa2* in their past. Ant, anterior; EB, early bud; LS, late-streak; nGFP, nuclear localised GFP; OB, no bud; Post, posterior; OFT, outflow tract; PS, primitive streak; RV, right ventricle. Scale bar: 100 μm.

notochord were enriched in tdTomato-positive cells (50% to 67%, scale data > 1, Fig 8Diii). Conversely, we found a lower frequency of tdTomato-positive cells or none in anterior mesoderm populations, including the anterior paraxial mesoderm (0%, scale data > 1). Among the cardiac progenitors, the pSHF had the highest proportion of tdTomato-expressing cells (43%, scale data > 1), followed by the AHF (10%, scale data > 1). The FHF had very few tdTomato-positive cells (3%, scale data > 1) (Fig 8Diii).

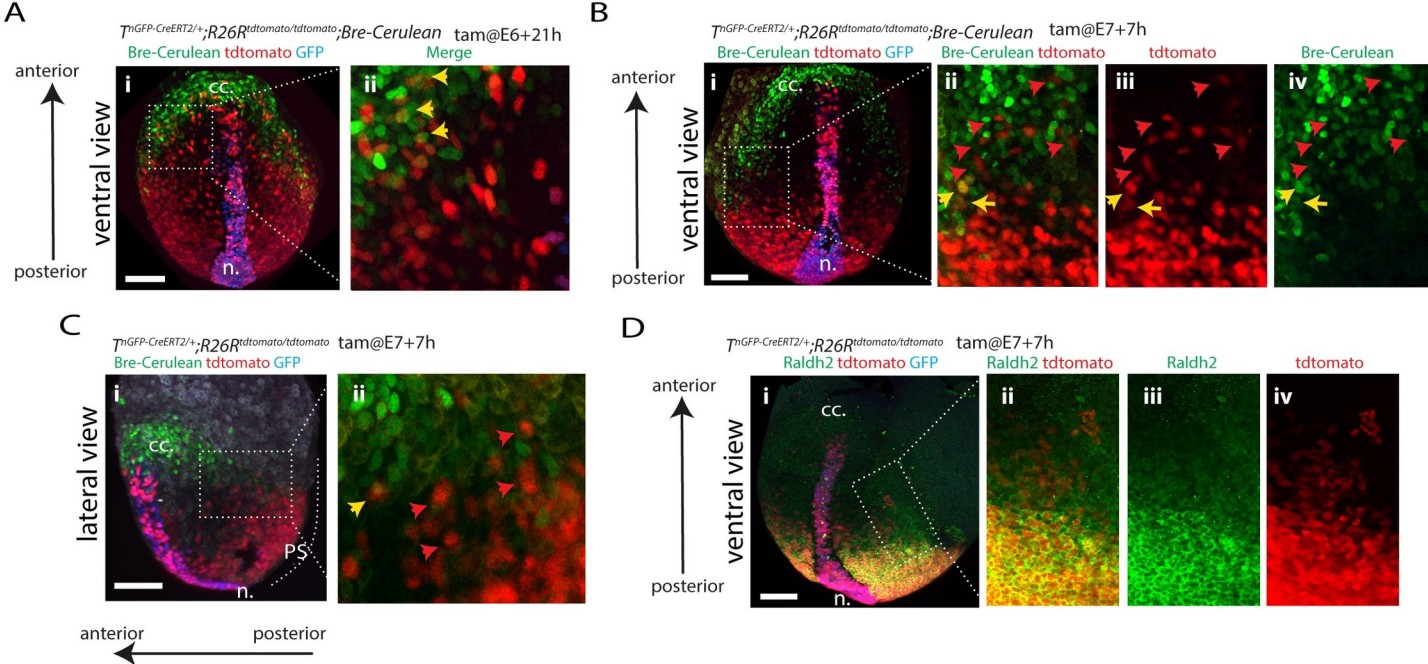

**Fig 7. Ventricular, outflow tract, and atrial progenitors are located in distinct regions of the mesoderm. (A, B)** tdTomato localisation in $T^{nGPF-CreERT2/+}$; $R26R^{tdTomato/tdTomato}$, *Bre-Cerulean* (BMP reporter) embryos at the EHF stage and following tamoxifen administration at E6+21h (A) and at E7+7h (B, C). Insets Aii, Bii–Biv, and Ci show magnified views of Ai, Bi, and Ci, respectively. Yellow arrows in Ai, Bii–Biv, and Cii point to double tdTomato/Bre-cerulean-positive cells and red arrows in Bii–Biv and Cii to tdTomato+ bre-cerulean− cells. **(D)** tdTomato localisation in $T^{nGPF-CreERT2/+}$; $R26R^{tdTomato/tdTomato}$ embryos at the EHF stage, following tamoxifen administration at E7+7h and immunostained for Raldh2. Insets in ii–iv show magnified view of i. Images are z-maximum projection of 37 sections acquired every 5 μm and covering 185 μm. Interval between frames: 6 mn and 30 s. cc, cardiac crescent; EHF, early head fold; ml, midline; n, node; PS, primitive streak. Scale bar: 100 μm.

Since the AHF contributes to both the right ventricle and the outflow tract [37], we next increased clustering's resolution to better distinguish tdTomato-positive from tdTomato-negative subpopulations. We observed the FHF, AHF, and pSHF clusters could be subdivided into 6 discernible clusters (Fig 8Biii, 8E and 8F, S13A Fig). Three of these clusters were enriched for tdTomato-positive cells (14%, 34%, and 74% for clusters 4, 5, and 6, respectively, scale data > 1). The 3 other clusters were depleted of tdTomato-positive cells (0%, 4%, and 3% for clusters 1, 2, and 3, respectively, scale data > 1). We hypothesised that the tdTomato-positive clusters included progenitors contributing to the outflow and atria. Conversely, we reasoned that the tdTomato-negative clusters comprised progenitors derived from earlier *T*-expressing cells contributing to the left and right ventricles and atrioventricular canal. We have summarised these results as a fate map for the different cardiac regions (Fig 11A and 11B). We propose that left and right ventricular, outflow, and atrial progenitors form molecularly distinct populations within the FHF, AHF, and pSHF at the EHF stage, mirroring their distinct origin in the primitive streak. We describe below the genes expressed in each of the tdTomato-positive (4, 5, and 6) and tdTomato-negative (1, 2, and 3) subpopulations in relation to the known expression patterns in cardiac progenitors (Fig 8F and 8G and S5 Source Data res12).

Cells from clusters 5 and 6—pSHF derived—expressed *Raldh2* and *Hoxb*1 (Fig 8C and 8F). Previous lineage tracing experiments have demonstrated that retinoic acid–activated cells contribute to the outflow and atria [38]. Additionally, *Hoxb1*-positive cells contribute to the atria and to the inferior wall of the outflow tract that subsequently forms the myocardium at the base of the pulmonary trunk, in addition to patches of myocardium in the ventricles [39–41]. Our data show *Tbx1*- and *Raldh2*-positive cells formed complementary cell populations within

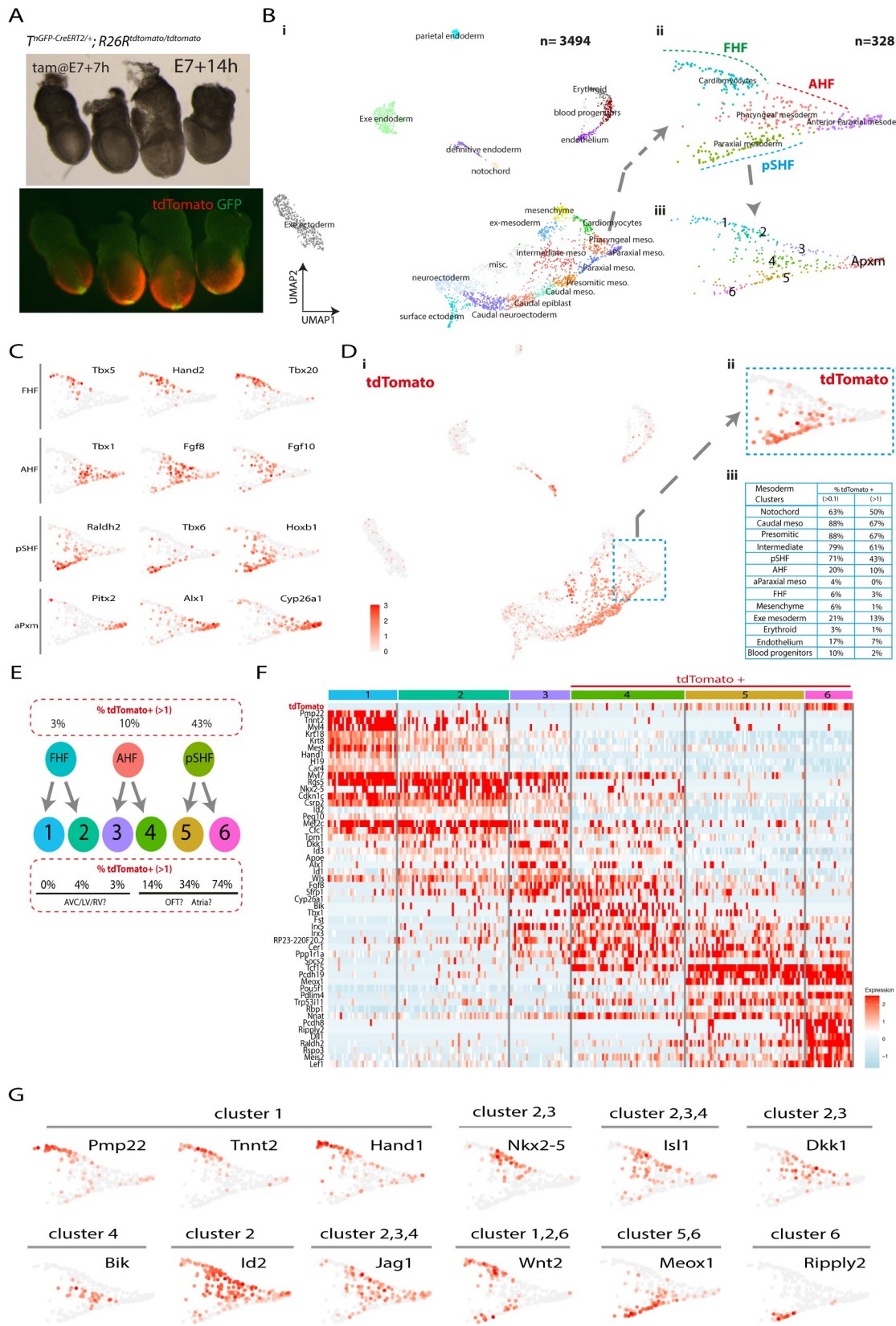

**Fig 8. Subclusters corresponding to the left and right ventricle, outflow tract, and atrial progenitors can be defined within the FHF, AHF, and pSHF.** (**A**) The 4 $T^{nGPF\text{-}CreERT2/+}$; $R26R^{tdTomato/tdTomato}$ embryos (dissected at E7+14 h EHF stage) analysed in the single-cell transcriptomic assay. (**B**) UMAP plot showing the cell populations coloured by cluster identity (i). (ii, iii) Magnified view of (i) showing the cardiomyocytes, pharyngeal mesoderm, anterior paraxial and paraxial mesoderm clusters (ii) and subclusters (iii). The cardiomyocytes, pharyngeal mesoderm, and paraxial mesoderm clusters are renamed FHF, AHF, and pSHF based on the marker genes shown in (C). (**C**) UMAP showing the log normalised counts of the selected genes. (**D**) UMAP showing the log normalised counts of the tdTomato gene (i). (ii) Magnified view of (i) showing the log normalised counts of the tdTomato gene in the FHF, AHF, and pSHF clusters. (iii) Table showing the percentage of tdTomato+ cells in each mesodermal cluster. The data underlying (iii) can be found in S6 Source Data. (**E**) Diagram showing the repartition of the tdTomato-positive cells from the FHF, AHF, and pSHF clusters into 6 subclusters, each with different proportion of tdTomato-positive cells (see also S13A Fig). We hypothesis that cells in clusters enriched in tdTomato reads contribute to the OFT and Atria. (**F**) Expression heat map of marker genes and tdTomato. Scale indicates z-scored expression values. The full heat map with all the genes display is shown S13B Fig. The data underlying (F) can be found in S5 Source Data. (**G**) UMAP showing the log normalised counts of selected genes. AHF, anterior second heart field; aParaxial meso, anterior paraxial mesoderm; AVC, atrioventricular canal; Caudal meso, caudal mesoderm; EHF, early head fold; Exe endoderm, extra-embryonic endoderm; FHF, first heart field; GFP, green fluorescence protein; Intermediate meso, intermediate mesoderm; misc, miscellaneous; n., node; OFT, outflow tract; Pharyngeal mes, Pharyngeal mesoderm; Presomitic meso, presomitic mesoderm; PS, Primitive streak; pSHF, posterior second heart field; Tam, tamoxifen; UMAP, Uniform Manifold Approximation Projection.

the tdTomato-positive clusters. *Tbx1* is expressed in clusters 4 and 5, and Raldh2 in clusters 5 and 6. This is consistent with lineage tracing experiments demonstrating that *Tbx1* expressing cells within the AHF contribute to the outflow tract [42]. We also found that *Wnt2* was expressed in clusters 1, 2, and 6. This is consistent with *Wnt2* expression in posterior meso-derm and the anterior mesoderm where the prospective cardiac crescent forms [4,43].

*Fgf8*, a known marker of the AHF [11], was enriched in both cluster 4 (low tdTomato posi-tive) and cluster 3 (tdTomato negative). Lineage tracing experiments have shown that *Tbx1* marks part of the right ventricle progenitor domain within the AHF [42,44] while *Fgf8* marks it entirely [36,45]. Consistent with the hypothesis that cluster 3 corresponded to a right ven-tricular progenitor domain, we found *Tbx1* to be enriched within a subset of cluster 3 (expressed in 30% of the cells). In comparison, *Fgf8* is expressed throughout (expressed in 70% of the cells). Also consistent with the hypothesis that clusters 2, 3, and 4 contained ventricular and outflow progenitors is the enrichment of *Jag1* in these 3 clusters, a gene shown to be expressed in progenitors derived from the *Foxa2* lineage [46]. *Jag1* is expressed anteriorly, complementary to the region where the cardiac crescent resides [47]. *Dkk1*, a Wnt modulator controlling cardiac differentiation [48,49], was also enriched in clusters 2 and 3. *Id 2*, a gene essential for the specification of the heart tube-forming progenitors [50] is also enriched in clusters 2 and 3.

## The ventricular and outflow tract myocardium are prepatterned within the primitive streak

The signalling environment that cells encounter during migration influences the patterning of the heart fields (S13B Fig) [12–14,30,46,51]. However, our findings that the ventricles and the outflow/atria arise from distinct primitive streak cell populations raise the possibility that these progenitors are already molecularly distinct in the primitive streak. To address this question, we analysed the transcripts of cells from tamoxifen-treated $T^{nEGP\text{-}CreERT2/+}$ $R26R^{tdTomato/tdTomato}$ embryos at the mid-late steak stages (3,635 cells from 9 embryos; Fig 9A) and OB-EB stages (3,994 cells from 4 embryos; Fig 9A). Because we administered tamoxifen in these embryos, tdTomato-positive mesodermal cells that had already emigrated could be visualised. In the mid-late streak stage embryos, tdTomato-positive cells were identified in the extraem-bryonic mesoderm, consistent with the idea that the earliest progenitors to emerge from the primitive streak are contributing to the yolk sac blood cells and vasculature endothelium [52]. tdTomato-positive cells were also sparsely found in the embryonic mesoderm. In the OB-EB stage embryos, tdTomato-positive cells were found in the embryonic mesoderm where

ventricular progenitors reside. Outflow and atrial progenitors are leaving the primitive streak at these stages. Unfortunately, the low level of expression at these times that meant very few tdTomato reads could be recovered and tdTomato-positive cells could therefore not be identified in our single-cell transcriptomic analysis at these early stages (S15A and S15B Fig).

We clustered cells guided by available atlases of mouse gastrulation [3] (Fig 9B and 9C and S14A–S14D Fig). We analysed cells corresponding to the primitive streak and aPS clusters at the mid-late streak stages and cells in primitive streak cluster at the OB-EB stages (Fig 9E). The primitive streak cells at the mid-late streak and OB-EB stages did not express the *Cdx1/4* TFs associated with a posterior mesodermal identity. Nor did these cells express *Raldh2*, suggesting that the outflow and atrial progenitors start expressing *Raldh2* once the cells have reached their final location in the embryo (S14E Fig).

Integration of the MS-LS and EB-OB datasets (Fig 9D) and differential gene expression analysis revealed major molecular differences between primitive streak cells contributing to either the ventricles or the outflow/atria (Fig 9E and 9F). The T-box transcription factor *Eomes* was strongly expressed in the mid-late streak stage primitive streak (ventricular progenitors) while its expression was lower at the OB-EB stages in the outflow/atrial progenitors (Fig 9G). This result is consistent with the function of *Eomes* in establishing the anterior mesoderm and specifying cardiac progenitors upstream of *Mesp1* [4]. *T* had the opposite pattern of expression and showed a stronger expression at the OB-EB stages compared to the mid-late streak stages (Fig 9G). Fate mapping in the mouse also identified the aPS adjacent to the organiser as a source of cardiac progenitors [52] and *Hes1*, *hHex* [53], *Gsc* [54], *Foxa2* [55], *Upp1*, *Tdgf1* [56,57], *Bmp2*, *Epha2*, *Lhx1* [58,59], *Otx2* [60], and *Chrd* [61] were among the genes enriched in the aPS at the mid-late streak stages (Fig 9E and 9H). *Sp8*, *Amt*, and *Notch2* [62] were preferentially enriched in the primitive streak at the OB-EB stages (Fig 6E and 6H). *Ccnd2* is expressed in the proximal regions of the primitive streak where the atrial progenitors are located [63] and where BMP signalling activity is high [64]. *Mixl1* [65], *Fst* [66], *Gas1* [67], *Frzb* [68], and *Apln* were expressed in the primitive streak at both the mid-late streak and OB-EB stages (Fig 9E). We conclude that ventricular, outflow, and atrial progenitors derive from molecularly distinct groups of cells that occupy spatially and temporally discrete regions of the primitive streak.

## Live imaging reveals the migratory routes of the mesodermal cells

Finally, to confirm that early and late *T*-expressing cells migrated to distinct anterior-posterior locations within the heart fields, we tracked single cell in live $T^{nEGP-CreERT2/+}$ $R26R^{mg/+}$ embryos (Fig 10A and S1 Movie). Early mesodermal cells from distal locations (yellow arrows in Fig 10A) migrated anteriorly along the midline to regions where the FHF is established. Conversely, both late distal and proximal mesodermal cells (orange and red arrows, respectively, in Fig 10A) contributed to more posterior regions of the mesoderm. The distal progenitors migrated to more medial regions (labelled as AHF, orange arrows in Fig 10A) and proximal cells to more lateral regions of the mesoderm (labelled as pCC and pSHF, red arrows in Fig 10E).

## Discussion

Our findings, summarised in Fig 11A–11C, reveal the temporal and spatial order in which different cardiac lineages arise within the primitive streak. The left ventricular progenitors are the first to leave the primitive streak at the mid-streak stage, followed shortly after by the right ventricular progenitors at the late-streak stage. Progenitors contributing to the poles of the heart leave the primitive streak at the OB-EB stages. The outflow progenitors arise from distal

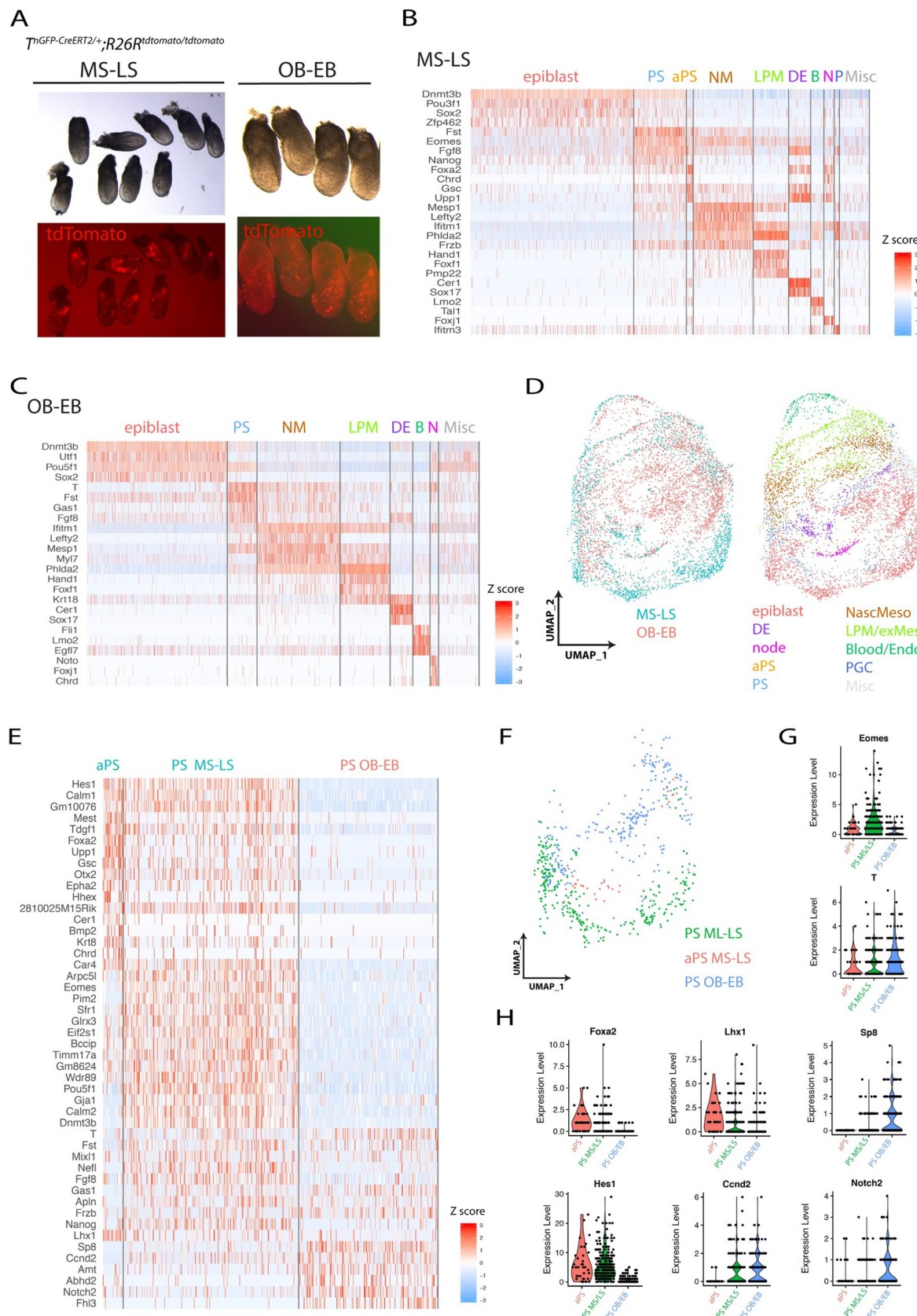

**Fig 9. The molecular signature of the primitive streak cells contributing to the ventricles or outflow/atria are distinct.** (**A**) Mid-late streak and OB/EB $T^{nGPF-CreERT2/+}$; $R26R^{tdTomato/tdTomato}$ embryos analysed in the scRNA-seq. Embryos were selected at the mid-late streak stages (dissected at E6+21h and resulting from tamoxifen administration at E6+5h) and at the OB-EB stage (dissected at E7+3h and resulting from tamoxifen administration at E6+21h). (**B, C**) Expression heat map of marker genes. Scale indicates z-scored expression values. The data can be found in S7 Source Data. (**D**) UMAP plot showing the integrated data from the 2 scRNA-seq mid-late streak and OB/EB datasets (see also S11 Fig). Colour codes correspond to the embryonic stage of collection or population identity. The data underlying (D) can be found in S7 Source Data. (**E**) Expression heat map of marker genes comparing the aPS and primitive streak cells at the mid-late streak stages and primitive streak cells at the OB-EB stages (S7 Source Data). Scale indicates z-scored expression values. (**F**) UMAP plot of the aPS, MS-LS primitive streak, and OB-EB primitive streak cells colour coded. (**G, H**) Violin plots showing the normalised log2 expression value of *Eomes* and *T* (G) and *Foxa2, Lhx1, Sp8 Hes1, Ccnd2,* and *Notch2* (H) in the aPS and primitive streak at MS-LS and OB-EB stages. The data underlying (G, H) can be found in S7 Source Data. aPS, anterior primitive streak; DE, definitive endoderm; EB, early bud; LPM/Ex-meso, lateral plate mesoderm and extraembryonic mesoderm, mesenchyme; LS, late streak; MS, mid-streak; Nascent meso, nascent mesoderm; OB, no bud; PGC, primordial germs cells; PS, primitive streak; scRNA-seq, single-cell RNA sequencing; UMAP, Uniform Manifold Approximation Projection.

regions of the primitive streak, while the atrial progenitors are located in proximal regions of the primitive streak. These different subpopulations constitute molecularly distinct groups of progenitors within the heart field and the transcriptional differences in the primitive streak cells suggest that cardiac progenitors are prepatterned in the primitive streak before their migration. This organisation of myocardial progenitors is conserved during evolution, and in

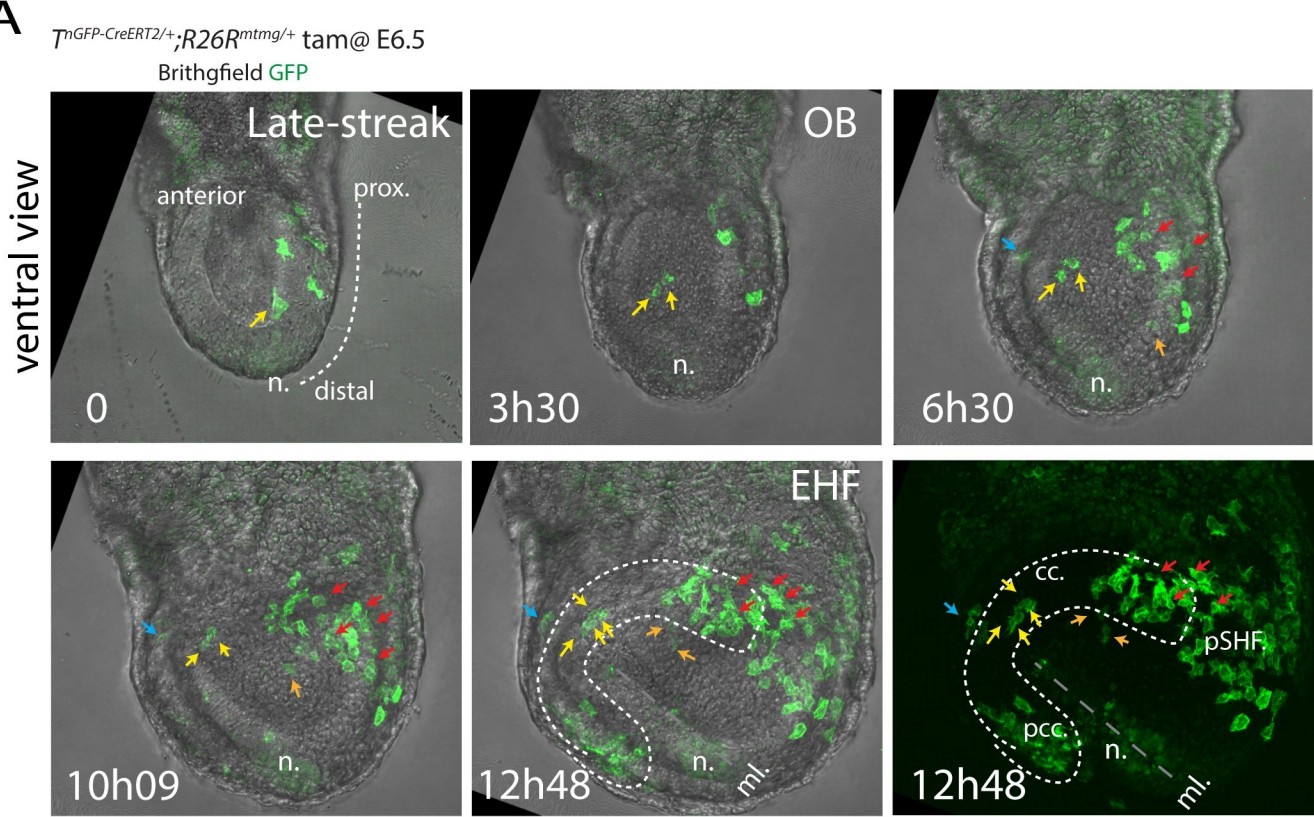

**Fig 10. Live imaging of the mesodermal cells reveal their trajectories during gastrulation.** (**A**) Image sequence from time-lapse video (S1 Movie) of an $T^{nGPF-CreERT2/+}$; $R26R^{mgpf/+}$ embryo resulting from the administration of tamoxifen at around E6.5 (overnight timed matings). Yellow arrows point to progenitor initially located in proximity to the node/distal regions and migrating along the midline in medial regions of the prospective cardiac crescent. Red arrows point to progenitors initially located proximally and migrated towards posterior regions of the prospective cardiac crescent and posterior second heart field. Blue arrow points to a progenitor located at the embryonic-extraembryonic border. Images are z-maximum projection of 37 sections acquired every 5 μm and covering 185 μm. Interval between frames: 6 mn and 30 s. cc, cardiac crescent; EHF, early head fold; ml, midline; n, node; OB, no bud; pCC, posterior cardiac crescent; pSHF, posterior second heart field. The movie can be found in S1 Movie.

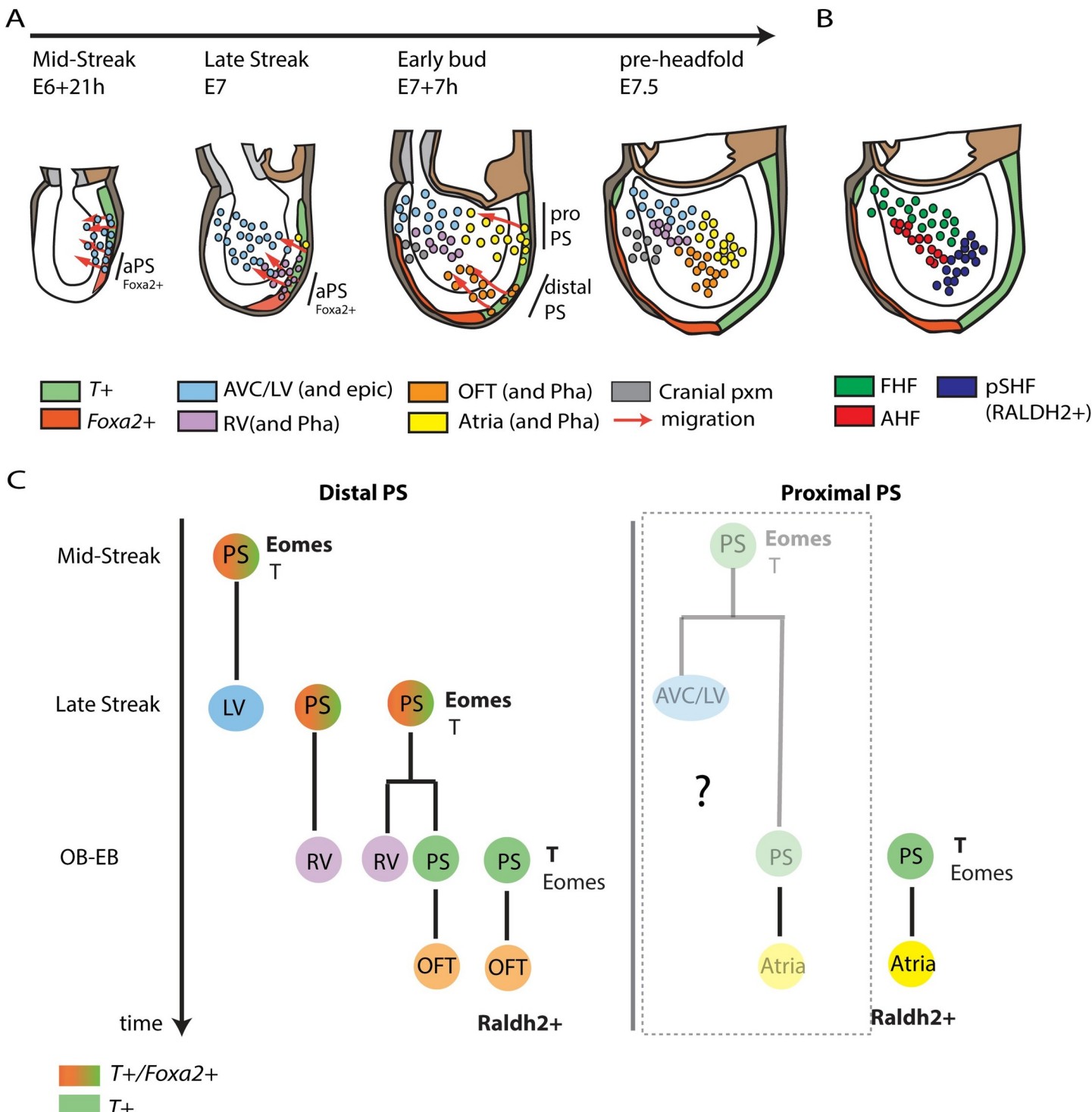

**Fig 11. Model of early cardiac development.** (**A**) Cells located in distal regions of the primitive streak contribute first to the left ventricle (mid-streak stage), then to the right ventricle (late-streak stage), and finally to the outflow tract (OB-EB stages). Although the outflow and atria leave the primitive streak at similar stages, they arise from different regions. The outflow tract originates from distal locations in the primitive streak while atrial progenitors are positioned more proximally. The distal primitive streak cells express *Foxa2* when they contribute to the ventricles. They stop expressing *Foxa2* when they contribute to the outflow tract. (**B**) Proposed location of the FHF, AHF, and pSHF in comparison to the hypothetical fate of these different cardiac regions at the pre-headfold stage in (A). (**C**) Lineage tree of the primitive streak. Within the distal end of the primitive streak, independent primitive streak cells contribute to the left ventricle, the right ventricle, the (at the mid-late streak stage) and to the outflow tract (OB-EB stages). A common progenitor between the right ventricle and the outflow tract exists. By contrast, atrial progenitors are located in the proximal primitive streak and a common progenitor between the left ventricle and atrium may exist in the primitive streak. AHF, anterior heart field; aPS, anterior primitive streak;

AVC, atrioventricular canal; cranial pxm, cranial paraxial mesoderm; EB, early bud; epi, epicardium; FHF, first heart field; LV, left ventricle; OB, no bud; OFT, outflow tract; Pha, Pharyngeal arches; PS, primitive streak; pSHF, posterior second heart field; RV, right ventricle.

zebrafish, the ventricular and atrial myocardial progenitors are spatially separate in the blastula [69].

### Primitive streak origin of the cardiac progenitors

While our *Foxa2* lineage tracing demonstrates an aPS origin of the ventricular myocardium, it is likely that more proximal regions of the PS marked by M*esp1* at the mid-late streak stage also contribute to the ventricles. Two lines of evidence support this hypothesis. First, the maximum contribution of the *Foxa2* progeny to the ventricles is 30% to 50% in *Foxa2*-based lineage tracing experiments ([8] and this study). This suggests that the *Foxa2* progeny represent a subset of the whole pool of ventricular progenitors. Second, fate mapping analysis in the mouse demonstrated that some cells from the middle/proximal regions of the PS also contributed to the cardiac crescent or FHF (which mostly form the left ventricle) in addition to the aPS [70]. Further investigation is required to characterise the descendants and contribution of the proximal primitive streak to the ventricles and whether or not they overlap with descendants of *Foxa2* progenitors. It is also unclear whether the *Foxa2* progeny contribute to a population of multipotent progenitors recently identified at the intra-/extraembryonic boundary in the FHF [24,25]. Instead, these progenitors may arise from an early population of proximal *Mesp1*-positive primitive streak cells (Fig 11C) [25].

Our *T*-lineage tracing experiments demonstrate that most of the atria arise from late OB-EB stages primitive streak. Our results further revealed that a subset of the atrial progenitors is gradually incorporated into posterior regions of the FHF to establish the inflow of the initial heart tube. This is consistent with the notion that the FHF contributes to part of the atria [21,22]. The earliest PS cells contribute to the most anterior regions of the FHF which are fated to give rise mostly to the left ventricle and atrioventricular canal in addition to the epicardium in the heart while a rare contribution to the dorsal aspect of the right atria is noted [25]. Thus, we propose that in the proximal regions of the primitive streak, progenitors contributing to the atrioventricular canal and left ventricle are the first to emigrate followed rapidly by the atrial progenitors and a subset of these will contribute to posterior regions of the FHF (Fig 11C).

### Lineage relationships between cardiac progenitors

The *Foxa2* lineage tracing demonstrates the existence of a common lineage between the right ventricle and the outflow. This suggests a population of cells resides within the PS, the progeny of which contributes successively to the right ventricle and outflow tract progenitor pools while self-renewing (see model in Fig 11A–11C). This is similar to a previous study that found the progeny of individually labelled primitive streak cells contributed to the notochord and somites while also leaving descendants in the primitive steak [71]. Live imaging experiments will provide a more direct indication of the existence of asymmetric cell divisions mediating this process within the primitive streak.

While the right ventricle and outflow tract progenitors were identified in the anterior/distal primitive streak, atrial progenitors were located in the proximal primitive streak. This spatial segregation suggests that atrial progenitors are likely to constitute a pool of progenitors independent from right ventricle and outflow cardiac progenitors because of their physical separation. This is consistent with the initial clonal analysis from Meilhac and colleagues [2], which

concluded that atrial cells became clonally distinct before the presumptive right ventricle and outflow tract [2]. Subsequent clonal experiments tracing *Mesp1*-positive progenitors [6,7] also resulted in small clones spanning both the right ventricle and outflow tract compartments, while independent clones were found in the atria [6,7] in addition to left ventricle–atrium clones [6]. Thus, atrial progenitors are likely to constitute a pool of progenitors in the primitive streak distinct from a common progenitor for the right ventricle and outflow tract (see the model in Fig 11C).

Meilhac and colleagues also observed 2 classes of larger clones extending across multiple compartments. This demonstrated the existence of 2 cardiac lineages in the embryo corresponding to the 2 heart fields [1,2]. Our analysis and others [6,7,25] suggest that the segregation of 2 cardiac lineages could only occur before the onset of gastrulation, for example, at the epiblast stage. A further subdivision into progenitors with more restricted cardiac lineages then arises when epiblast cells ingress through the primitive streak during gastrulation. This is reflected by the existence of progenitor subpopulations within the FHF and SHF corresponding to the prospective left and right ventricle, outflow tract, and atria.

## A combination of intrinsic factors and inductive events specify and pattern the cardiac progenitors

Depending on the initial timing and site of ingression through the primitive streak, cardiac progenitors might adopt distinct migratory routes that expose them to different signalling environments and influencing their cardiac fate. BMPs and Fgfs secreted by the anterior endoderm [72–75], retinoic acid expressed in the posterior lateral plate mesoderm [14,15], and signals expressed in the anterior intestinal portal (AIP) at later stages [76] all contribute to the patterning of the cardiac progenitors. A Hedgehog-Fgf signalling axis has also been proposed to pattern the anterior mesoderm to allocate the head and heart during mesodermal migration [13]. Our analysis shows that the primitive streak cells that contribute to either the ventricles or the poles of the heart are molecularly distinct prior to migration. For example, changes in the transcriptional profile of primitive streak cells, including the down-regulation of *Foxa2* expression, accompany the transition from contributing to the right ventricle to supplying the outflow tract. This mode of cellular diversification is reminiscent of temporal programmes in other tissues, such as neural progenitors [77], where sequential expression of distinct sets of transcription factors in progenitors defines the later differentiated cell types. These findings raise the possibility that cardiac progenitors are specified via a combination of both initial conditions set in the primitive streak and inductive events happening during migration [13,46] or once cells have reached their final location in the embryo.

Our analysis could not resolve putative differences between left and right ventricular progenitors in the primitive streak. A more precise staging of embryos separating the mid from the late streak stages within a single-cell transcriptomic assay will be required for this. Transcriptional differences between left and right ventricular progenitors may be established during migration by exposure to BMP and FGF signalling, both of which are known to mediate mediolateral patterning of the mesoderm [36,37,78,79].

The observation that initial molecular differences in the primitive streak lead to mesodermal cells that generate distinct populations of cardiac progenitors might inform in vitro cardiac differentiation protocols [80,81]. We show that atrial cells are located in the proximal primitive streak where BMP signalling marked by P-Smad1/5/8 is high [64]. By contrast, ventricular progenitors originate in aPS regions where cells are exposed to high Nodal signalling. This is consistent with methods to generate cardiac cells from human ESCs in vitro that rely on exposure to different levels of Nodal and BMP signalling [80]. A higher ratio of BMP4 to

activin A signalling is required for the generation of Raldh2-positive mesoderm, which forms atrial cells. Conversely, the generation of ventricular cardiomyocytes relies on a higher level of activin to BMP4 signalling and the formation of Raldh2-negative CD235a-positive mesoderm [80].

We conclude that cardiac progenitors are prepatterned within the primitive streak, and this prefigures their allocation to distinct anatomical structures of the heart. Further work will be required to test if generating the correct population of primitive streak cells can obtain purer populations of cardiomyocytes from pluripotent stem cells. Our analysis will also help identify the initial mesodermal population that when dysregulated leads to specific malformations in the heart. For example, left ventricle hypoplasia results from a reduction in the number of specified cardiomyocytes within the mesoderm, and this is due to a reduction in the expression of key cardiac transcription factors [82]. Whether such cardiac defects can arise from mutations affecting initial primitive streak cell populations and mesoderm remains to be addressed.

## Methods

### Experimental model and subject details

All animal procedures were performed in accordance with the Animal (Scientific Procedures) Act 1986 under the UK Home Office project licenses PF59163DB and PIL IA66C8062.

### Mice

The $T^{nEGFP-CreERT2/+}$ (MGI:5490031) and $Foxa2^{nEGP-CreERT2/+}$ (MGI:5490029) lines [16] were obtained from Hiroshi Sasaki. The $R26^{Tomato\ Ai14/\ Tomato\ Ai14}$ ($Gt(ROSA)26Sor^{tm14(CAG-tdTomato)Hze}$ (MGI:3809524), $Gt(ROSA)26Sor^{tm4(ACTB-tdTomato,-EGFP)Luo,}$ (MGI: 3716464) were obtained from the Jackson Laboratory. The $Mesp1^{tm2(cre)Ysa}$ (MGI:2176467) line was obtained from MRC Harwell, Mary Lyon. The TCF/Lef:H2B:mCherry WNT reporter line was generated in house by the Briscoe laboratory [83].

### BRE:H2B-Turquoise BMP reporter line

The MLP:H2B-Turquoise sequence (originally from BRE-MLP-H2B-Turquoise pGL3 basic, Briscoe lab, Addgene, ID number: 171499) was cloned using In-Fusion Cloning (Takara, 639650) between 2 chicken insulators at the BamHI site of plasmid pJC5-4 [84] without the LCR fragment. The 1.6-kb MLP:H2B-Turquoise fragment was amplified using Phusion High-Fidelity PCR Master Mix (ThermoFisher Scientific, F532L) according to the manufacturer's instructions. Subsequently, the 92-bp BRE element was isolated with MluI/XhoI digest from BRE-MLP-H2B-Turquoise pGL3 basic and cloned 5′ to the MLP sequence of MluI/XhoI-digested MLP:H2B-Turquoise pJC4-4 plasmid. The final plasmid was linearized with NdeI and used for pronuclear injection using fertilised embryos from the F1 hybrid strain (C57BL6/CBA). Mice with BRE reporter activity were verified by genotyping by a commercial vendor (Transnetyx). 5′-3′ Forward primer: CACAAGCTGGAGTACAACTACATCAGCGA Reporter 1: TCTATATCACCGCCGAC Reverse Primer: GGCGGATCTTGAAGTTGG CCTTGA. The BRE reporter line was maintained on the F1 background by crossing heterozygous *BRE:H2B-Turquoise* mice to F1 wild-type mice.

### Lineage tracing of the *T*- and *Foxa2*-expressing cells

Lineage tracing of *T and Foxa2* mesodermal progenitors was performed by crossing $T^{nEGFP-CreERT2/+}$ and $Foxa2^{nEGP-CreERT2/+}$ with $R26^{Tomato\ Ai14/\ Tomato\ Ai14}$ mice. To gain better control over embryonic staging, mice were synchronised in estrus by introducing soiled bedding from

a male's cage into the females' 3 days in advance. On the fourth day, mice were crossed over a 2-hour period from 7 AM to 9 AM. Vaginal plugs were checked at 9 AM and, if positive, the embryonic day was defined at E0. This was followed by tamoxifen oral gavage with tamoxifen at 0.08 mg/body weight (T5648 SIGMA) dissolved in corn oil at indicated times.

## Whole-mount immunofluorescence and image acquisition

Embryos were fixed for 20 minutes (for early E6.5-E7.5 embryos) or overnight (for late E12.5 hearts) in 2% PFA at 4°C, then permeabilized in PBST (PBS containing 0.5% Triton X-100) for 15 minutes (for early E6.5-E7.5 embryos) or 1 hour (for late E12.5 hearts) and blocked for 5 hours (5% donkey serum, Abcam ab138579). Embryos were incubated overnight at 4°C with antibodies diluted in PBST (PBS 0.1% Triton X-100): rabbit anti-Oestrogen Receptor alpha antibody (Sp1) (1:100, Abcam ab16660), rat anti-Flk1 (1:250, BD Biosciences 55307), goat anti-DKK1 (R&D Systems, AF1765), rabbit ranti-Aldh1a2 (Abcam ab96060), rat anti-CER1 (R&D Systems, MAB1986), goat anti-T (1:250, R&D Systems AF2085), rabbit anti-FOXA2 (1:250, Abcam ab108422); mouse anti-cTNT2 (1:250, Thermo Fischer Scientific Systems, MS295P0); rabbit anti-PhosphoSMAD1/5/8 (1:250, Cell Signalling D5B10 Rabbit mAb #13820). After washing in freshly prepared PBST at 4°C, embryos were incubated with secondary antibodies (Molecular Probes and Biotium) coupled to AlexaFluor 488 or 647 fluorophores and CF430 as required at 1:250 overnight at 4°C. Before imaging, embryos were washed in PBST at room temperature. Confocal images were obtained on an inverted Zeiss 710 confocal microscope with a 20X water objective (for early E6.5-E7.5 embryos) or a 10X air objective (0.4 NA) (for late E12.5 hearts) at a 1024 × 1024 pixels dimension with a z-step of 3 to 6 μm (2 × 2 tile scale, for the late E12.5 hearts). Embryos were systematically imaged throughout from top to bottom. Images were processed using Fiji software [85].

## Image analysis

To segment the Foxa2- and T-positive cells, a Gaussian filter whose radius is adjusted to the typical size of a cell was first applied with Fiji [85]. The resulting image was next converted to a mask by thresholding. When objects touched each other, a watershed on the binary mask. Finally, particle analyser generated a binary image with objects outside specified size (50) and circularity (0.75 to 1.00) removed. This process was repeated in each optical z-section of the z-stack. To quantify the surface area of the tdTomato-labelled myocardial cells, patches of tdTomato cells were manually segmented and area measured with Fiji [85] for each individual heart within each litter.

## PCR analysis

Primers were designed to span the floxed SV40 poly(A) signal sequence removed from the genome following Cre recombinase-mediated recombination: 5'-3' Forward primer: CGTGCTGGTTATTGTGCTGT; Reverse: CATGAACTCTTTGATGACCTCCTCGC. Primers yield a 1,145-bp product from unrecombined DNA and a 274-bp product following recombination. gDNA was extracted using the HotSHOT method [86] from either the ear clip of an adult $R26^{Tomato\ Ai14/\ Tomato\ Ai14}$ mouse (unrecombined) or tail bud dissected from E9.5 embryos from a pregnant $T^{nEGFP-CreERT2/+}$ $R26^{Tomato\ Ai14/\ Tomato\ Ai14}$ mouse, at 2, 4, and 12 hours following oral gavage with Tamoxifen at 0.08 mg/body weight (T5648 SIGMA) dissolved in corn oil. gDNA was then used in Q5 High-Fidelity 2X Master Mix (NEB) PCR reactions according to the manufacturer's instructions. Amplicons were resolved on a 2% agarose gel with a 100-bp ladder.

## Sample preparation for single-cell RNA sequencing

$T^{nEGFP\text{-}CreERT2/+} R26^{Tomato\ Ai14/Tomato\ Ai14}$ mouse embryos were imaged prior to dissociation with a Leica stereo fluorescent microscope with an exposure of approximately 1 mn to reveal the tdTomato signal. Sample preparation was done using previously established method [87]. Mouse embryos were dissected in Hanks Balanced Solution without calcium and magnesium (HBSS, Life Technologies, 14185045) supplemented with 5% heat-inactivated foetal bovine serum (FBS). The samples were then incubated on FACSmax cell dissociation solution (Amsbio, T200100) with 10× Papain (30 U/mg, Sigma-Aldrich, 10108014001) for 11 minutes at 37˚C to dissociate the cells. To generate a single-cell suspension, samples were transferred to HBSS, with 5% FBS, rock inhibitor (10 μM, Stemcell Technologies, Y-27632) and 1× nonessential amino acids (Thermo Fisher Scientific, 11140035), disaggregated through pipetting, and filtered once through 0.35 μm filters and once through 0.20 μm strainers (Miltenyi Biotech, 130-101-812). Quality control was assayed by measuring live cells versus cell death, cell size, and number of clumps, and 10,000 cells per sample were loaded for sequencing.

## Analysis of scRNA-seq data

A suspension of 10,000 single cells was loaded onto the 10x Genomics Single Cell 3′ Chip, and cDNA synthesis and library construction were performed as per the manufacturer's protocol for the Chromium Single Cell 3′ v2 protocol (10x Genomics; PN-120233) and sequenced on an Hiseq4000 (Illumina). 10x CellRanger (version 3.0.2) was used to generate single-cell count data for each sample using a custom transcriptome built from the Ensembl mouse GRCm38 release 86 with the addition of the sequence from tdTomato Ai9 plasmid [17]. Due to 10x's poly A bias, an additional 225 bases between the tdTomato gene stop codon and the bGH poly (A) signal was included to represent the WPRE gene. Depth of sequencing was 47,975 mean reads per cell for the E7+14h dataset, 32,629 mean reads per cell for the MS-LS dataset, and 64,340 mean read per cell for the OB-EB dataset. All subsequent analyses were performed with R (v.3.6.1) (R Core Team (2013)) using the Seurat (v3) package [88]. Primary filtering was performed on each dataset by removing from consideration: cells expressing unique feature counts fewer than 500, number of Unique Molecular Identifiers (UMIs) fewer than 1,000, and cells for which mitochondrial genes made up greater than 3 times the standard deviation value of all expressed genes. Each dataset was normalised using the "LogNormalize" function, with a scale factor of 10,000. The top 2,000 highly variable genes were found using the "FindVariableGenes" function and the data centred and scaled using the "ScaleData" function. PCA decomposition was performed and after consideration of the eigenvalue "elbow-plots," the first 50 components were used to construct Uniform Manifold Approximation Projection (UMAP) plots. Samples E7+14h and E7.75 from Pijuan and colleagues [3] were integrated using Seurat standard integration workflow. Common anchor features between both experiments were selected using the "FindIntegrationAnchors" functions, using the first 20 dimensions. Cluster identity from Pijuan and colleagues E7.75 dataset [3] was next transferred to the E7+14h dataset. Cluster specific gene markers were identified using the "FindMarkers" function using the settings (min.pct = 0.25; min.diff.pct = 0.1; and return.thresh = 0.0001), which uses the Wilcoxon rank sum test to compare each cell belonging on one cluster versus all other cells. Genes were ranked based on logFC and the highest logFC was used to determine unique markers per cluster. Selected cluster marker genes were used to draw a heatmap showing the expression of tdTomato on the top row (S12A Fig and S5 Source Data). The clustering analysis was then repeated by increasing the resolution parameter and discernible clusters with marker genes corresponding to clusters 1 to 6 and anterior paraxial mesoderm were identified (S5 Source Data). Cluster specific markers were also identified, after subsetting only clusters, 1, 2, 3, 4, 5,

and 6 (Fig 8F and S5 Source Data). Samples E6+21h and E7+3h were similarly integrated using Seurat standard integration workflow. Common anchors feature between both experiments were selected using the "FindIntegrationAnchors" functions, using the first 20 dimensions. Clusters were labelled and grouped using preexisting cell markers [3] and differential expression between clusters were determined using the "FindMarkers Function," without logfc or minimum percentage expressed thresholds. Genes showing the largest logFC and smallest adjusted $p$-value between clusters were used to generate heatmaps (S7 Source Data).

## Embryo culture and two-photon live imaging

Embryos were dissected at E6.5 in preequilibrated DMEM supplemented with 10% FBS, 25 mM HEPES-NaOH (pH 7.2), penicillin (50 µml21), and streptomycin (50 mgml21). Embryos were cultured in 100% fresh rat serum filter sterilised through a 0.2-mm filter. To hold embryos in position during time-lapse acquisition, we made bespoke plastic holders with holes of different diameters (0.3 to 05 mm) to ensure a good fit of embryos similarly to the traps developed by Ivanovitch and colleagues [89] and Nonaka and colleagues [90,91]. Embryos were mounted with their anterior side facing up. To avoid evaporation, the medium was covered with mineral oil (Sigma-Aldrich; M8410). Before starting the time-lapse acquisition, embryos were precultured for at least 2 hours in the microscopy culture set up. The morphology of the embryo was then carefully monitored and if the embryos appeared unhealthy or rotated and or moved, they were discarded, otherwise, time-lapse acquisition was performed. For the acquisition, we used the multiphoton MPSP5 equipped with a 5% CO2 incubator and a heating chamber maintaining 37˚C. The objective lens used was a HCX APO L 20x/1.00 W dipping objective, which allowed a 2-mm working distance for imaging mouse embryos. A SpectraPhysics MaiTai DeepSee pulsed laser was set at 880 nm and used for one-channel two-photon imaging. Leica Las AF software was used for acquisition. Image settings was: output power: 250 mW, pixel dwell time: 7 µs, line averaging: two and image dimension: $610 \times 610$ µm ($1,024 \times 1,024$ pixels). To maximise the chance of covering the entire embryo during the long-term time-lapse video, we allowed 150 to 200 µm of free space between the objective and the embryo at the beginning of the recording.

## Supporting information

**S1 Fig. Tamoxifen activity persists for at least 24 hours when administrated at a high dose by oral gavage. (A)** The administration of a high dose of tamoxifen (0.08 mg/bw by oral gavage) at E5 in $T^{nGPF-CreERT2/+}$; $R26R^{tdTomato/+}$ mice leads to the presence of tdTomato-positive cells in mesoderm derivatives including cardiomyocytes (see yellow arrow in inset), head mesenchyme (red arrow), endothelium (green arrow), and allantois. a, allantois; hm, head mesoderm; ht, heart tube. Mouse were mated for a 2-hour period. Scale bar: 100 µm. (TIF)

**S2 Fig. Genetic tracing of the *T*+ primitive steak cell with the R26mt/mg reporter. (A)** Representative hearts resulting from the administration of tamoxifen at E6+8h (i) and E7+7h (ii, iii) in $T^{nGPF-CreERT2/+}$; $R26R^{mtmg/+}$ immunostained with cTnnT to reveal the cardiomyocytes (blue). (TIF)

**S3 Fig. CreErt2 nuclear localisation 2 hours after tamoxifen administration. (A)** Representative embryos resulting from a 2-hour pulse of tamoxifen via oral gavage (0.08 mg/bw) immunostained with oestrogen receptor. Embryos have been immunostained simultaneously and image under the same conditions. Maximum z-projection (i–iii) and single optical sections

(iv–vi) are shown.
(TIF)

**S4 Fig. Recombination of the R26RtdTomato reporter is occurring 2.5 hours after tamoxifen administration by oral gavage. (A, B)** PCR amplicons generated from the genomic region in which Cre-mediated recombination occurs from $T^{nGPF-CreERT2/+}$; $R26R^{tdTomato/tdTomato}$ s embryos (A), resolved on an agarose gel (B). Before recombination, the PCR product is 1,145 bp (white rectangle); after recombination, it is 274 bp (black rectangle). Template gDNA was extracted from either an ear clip of an adult $T^{nGPF-CreERT2/+}$; $R26R^{tdTomato/tdTomato}$ mouse (untreated) or $T^{nGPF-CreERT2/+}$; $R26R^{tdTomato/tdTomato}$ embryos (i, ii) following oral gavage with Tamoxifen, as labelled. An increase in the proportion of the recombined band can be seen over time following Tamoxifen administration. The data can be found in S2 Raw image.
(TIF)

**S5 Fig. *Foxa2*-expressing cells contribute to the outflow tract myocardium but not to the atria. (A)** Heart resulting from the administration of tamoxifen at E6+21h. View is ventral. tdTomato-positive cardiomyocytes are absent from the myocardium in the atria; however, contribution to the epicardium (yellow arrow) and myocardium (yellow arrows) in the ventricle and outflow tract is visible. (**B**) E8 embryo resulting from the administration of tamoxifen at E6+21h in $Foxa2^{nGPF-CreERT2/+}$; $R26R^{tdTomato/+}$ mouse and immunostained for Foxa2 (green). tdTomato-positive cells are localised in the pericardium, cardiomyocytes and endoderm but not in the endocardium. cardio, cardiomyocyte; CC, cardiac crescent; cTnnT, cardiac troponinin T; endo, endoderm; LA, left atria; LV, left ventricle; OFT, outflow tract; RA, right atria; RV, right ventricle. Scale bars: 200 μm in (A) and 100 μm in (B).
(TIF)

**S6 Fig. T and Foxa2 colocalise in primitive streak cells. (A–D)** Single optical sections from same embryos as shown in Fig 4. E6+21h MS (A–A") and LS (B–B") and E7+7h EB (E, F) embryos are immunostained for T (red) and Foxa2 (green). Views are lateral/slightly posterior. Insets in Ai, Aii, Bi, and Di show magnified views (A–C). White arrows point to T+/Foxa2+ double positive cells in the definitive endoderm (Ai, Aii) at MS position in MS-LS embryos (Bi and Ci). Scale bar: 100 μm. EB, "early bud" stage; LS, late-streak; MS, mid-streak; PS, primitive streak.
(TIF)

**S7 Fig. Segmentation of the proximal and distal primitive streak cells. (A)** Example of a segmented images based on T signal for the proximal cells (i) and Foxa2 signal for the distal cells (ii). Segmentation for only the tdTomato-positive cells is shown in (iii). Merge of the 2 segmented images (i and ii) is shown in (iv).
(TIF)

**S8 Fig. Characterisation of the *Foxa2* lineage-positive mesodermal cells. (A)** Representative $Mesp1^{cre/+}$; $R26R^{mGFP/+}$ embryo at about E7.5. (**B, C**) Representative embryos resulting from the administration of tamoxifen at E6+21h in $Foxa2^{nGPF-CreERT2/+}$; $R26R^{tdTomato/+}$ immunostained for Foxa2 (blue) and Cer1 (blue) (B) or Foxa2 (blue) and Flk1 (green) (C). Inset in Bi–Ci show magnified view (B, C) in single optical section. (**D**) Representative E7.5 embryo immunostained for DKK1 (red) and Foxa2 (green). (**E**) Representative TCFdsred embryo (red) at E7.5 immunostained for Foxa2 (green). Ant, Anterior; post., posterior; PS, primitive streak. Scale bar: 100 μm.
(TIF)

**S9 Fig. Bre-cerulean line report BMP signalling activity in the mesoderm.** (**A–A"**) Colocalisation of the Cerulean signal and P-Smad1/5/8 in Bre-cerulean embryos at the cardiac crescent stage. (A) z-max proj. (A') Single optical projection. (A") Magnified view form insets in A'. cc., cardiac crescent; e, endoderm; p, pericardium. Scale bar: 100 μm.
(TIF)

**S10 Fig. Outflow tract and atrial progenitors are located away from regions with high BMP signalling activity.** (**A**) tdTomato localisation in $T^{nGPF-CreERT2/+}$; $R26R^{tdTomato/tdTomato}$ embryos immunostained against P-Smad1/5/8 following tamoxifen administration at E7+7h. cc, cardiac crescent; pm, pharyngeal mesoderm. Yellow arrow points to a Phospho-Smad1/5/8 +/tdTomato+ cell, red arrows point to Smad1/5/8−/tdTomato+ cells. Scale bar: 100 μm.
(TIF)

**S11 Fig. Assignment of cluster identities in scRNA-seq E7+7h dataset.** (**A**) UMAP plot of the Pijuan and colleagues E7.75 dataset [3]. (**B**) UMAP plot of the E7+7h $T^{nGPF-CreERT2/+}$; $R26R^{tdTomato/tdTomato}$ dataset clustered at resolution 4.5. (**C, D**) UMAP plot showing the integrated data from the 2 scRNA-seq E7+7h $T^{nGPF-CreERT2/+}$; $R26R^{tdTomato/tdTomato}$ and Pijuan and colleagues E7.75 dataset [3]. Colour codes correspond to the embryonic stage of collection or population identity (**C**) and clusters (**D**). Note, the paraxial mesoderm cluster is split into 2 subclusters we named "paraxial mesoderm" and "anterior paraxial mesoderm" based on expression of marker genes (see also Fig 8D, S12A Fig, and S5 Source Data).
(TIF)

**S12 Fig. scRNA-seq analysis of the E7+7h $T^{nGPF-CreERT2/+}$; $R26R^{tdTomato/tdTomato}$ dataset.** (**A**) Expression heat map of marker genes (S5 Source Data) and tdTomato. Scale indicates z-scored expression values. (**B**) UMAP showing the log normalised counts of selected genes (**C**) Percentage of tdTomato-positive cells in each cluster for expression values above 0.1 and 1. The data underlying (C) can be found in S6 Source Data. (**D**) Dot plot of factors with restricted expression in progenitors. Dot size corresponds to the percentage of cells expressing the feature in each cluster, while the colour represents the average expression level. The data underlying (D) can be found in S5 Source Data. aPxm, anterior paraxial mesoderm; cm, cardiomyocytes; Phm, pharyngeal mesoderm; Pxm, paraxial mesoderm.
(TIF)

**S13 Fig. Identification of mesodermal subclusters and gradient of tdTomato expression.** (**A**) Repartition of the cells from the FHF, AHF, pSHF, and aPxm cluster to subclusters 1, 2, 3, 4, 5, and 6 and aPxm. The data underlying (A) can be found in S6 Source Data. (**B**) Dot plot of factors with restricted expression in progenitors. Dot size corresponds to the percentage of cells expressing the feature in each cluster, while the colour represents the average expression level. The data underlying (B) can be found in S5 Source Data. AHF, anterior heart field; aPxm, anteriorparaxial mesoderm; FHF, first heart field; pSHF, posterior second heart field.
(JPG)

**S14 Fig. scRNA-seq analysis of the mid-late streak and OB-EB stages.** (**A** and **C**) UMAP plot coloured by cluster identity from scRNA-seq analysis of $T^{nGPF-CreERT2/+}$; $R26R^{tdTomato/tdTomato}$ embryos at the E6+21h, MS-LS stages (A) and at the E7+3h, OB-EB stages (C). (**B** and **D**) UMAP showing the log normalised counts of selected genes. Colour intensity is proportional to the expression level of a given gene. aPS, anterior primitive streak; DE, definitive endoderm; EB, early bud; LPM/Ex-meso, lateral plate mesoderm and extraembryonic mesoderm; mesenchyme; LS, late streak; MS, mid-streak; Nascent meso, nascent mesoderm; OB, no bud; PGC, primordial germs cells; PS, primitive streak; scRNA-seq, single-cell RNA sequencing; UMAP,

Uniform Manifold Approximation Projection.
(TIF)

**S15 Fig. tdTomato reads in $T^{nGPF\text{-}CreERT2/+}$; $R26R^{tdTomato/tdTomato}$ at the mid-late streak and OB-EB stages. (A, B)** Violin plots showing tdTomato expression for each cluster in $T^{nGPF\text{-}CreERT2/+}$; $R26R^{tdTomato/tdTomato}$ mid-late streak (A) and OB-EB (B) embryos shown in Fig 9A. The data underlying (A, B) can be found in S7 Source Data.
(TIF)

**S1 Raw image. Related to Fig 1G.**
(TIF)

**S2 Raw image. Related to S4 Fig.**
(TIF)

**S1 Source Data. Quantification of labelled surface area related to Fig 1D and 1E.**
(XLSX)

**S2 Source Data. Quantification of T and ERT intensities related to Fig 1K and 1M.**
(XLSX)

**S3 Source Data. Quantification of labelled surface area related to Fig 3D and 3E.**
(XLSX)

**S4 Source Data. Quantification of T and Foxa2 intensities related to Fig 5E and 5F.**
(XLSX)

**S5 Source Data. Single-cell RNA sequencing data related to Fig 8F, S12A and S12D and S13B Figs.**
(XLSX)

**S6 Source Data. Quantification of the tdTomato-positive cells in each cluster related to Fig 8D–8F and S13A Fig.**
(XLSX)

**S7 Source Data. Single-cell RNA sequencing data related to Fig 9, S14A–S14D and S15A and S15B Figs.**
(XLSX)

**S1 Movie. Related to Fig 10.**
(AVI)

## Acknowledgments

The authors would like to thank the Science Technology Platforms at the Francis Crick Institute. In particular, we thank the Advanced Light Microscopy facility, the Advanced Sequencing Facility, Bioinformatics and Biostatistics Facility, and the Biological Research Facility for their ongoing support and access to equipment. We are grateful to Robert Goldstone and Amelia Edwards for excellent support with single-cell sequencing, Teresa Rayon for assistance with single-cell preparation, Gavin Kelly for support with single-cell data processing, and Joe Brock for research illustration. We thank Teresa Rayon, Florencia Cavodeassi, and Peter Scambler for comments on the manuscript and members of the Smith and Briscoe lab for useful discussion.

## Author Contributions

**Conceptualization:** Kenzo Ivanovitch, James Briscoe.

**Data curation:** Julien Delile.

**Formal analysis:** Kenzo Ivanovitch, Pablo Soro-Barrio, Probir Chakravarty.

**Funding acquisition:** James C. Smith, James Briscoe.

**Investigation:** Kenzo Ivanovitch, Rebecca A. Jones, S. Neda Mousavy Gharavy, Despina Stamataki.

**Methodology:** Donald M. Bell.

**Supervision:** James C. Smith, James Briscoe.

**Visualization:** Kenzo Ivanovitch, Rebecca A. Jones.

**Writing – original draft:** Kenzo Ivanovitch.

**Writing – review & editing:** James C. Smith, James Briscoe.

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
