## [Editor Report · Decision Letter 0]

14 Jul 2020

Dear Dr Ivanovitch, 

Thank you for submitting your manuscript entitled "Ventricular, atrial and outflow tract heart progenitors arise from spatially and molecularly distinct regions of the primitive streak." for consideration as a Research Article by PLOS Biology.

Your manuscript has now been evaluated by the PLOS Biology editorial staff as well as by an academic editor with relevant expertise and I am writing to let you know that we would like to send your submission out for external peer review.

Please re-submit your manuscript within two working days, i.e. by Jul 16 2020 11:59PM.

Kind regards,

Di Jiang, PhD,

Senior Editor

PLOS Biology

---

## [Decision Letter · Decision Letter 1]

14 Aug 2020

Dear Dr Ivanovitch,

Thank you very much for submitting your manuscript "Ventricular, atrial and outflow tract heart progenitors arise from spatially and molecularly distinct regions of the primitive streak." for consideration as a Research Article at PLOS Biology. Your manuscript has been evaluated by the PLOS Biology editors, an Academic Editor with relevant expertise, and by several independent reviewers.

In light of the reviews (below), we will not be able to accept the current version of the manuscript, but we would welcome re-submission of a much-revised version that takes into account the reviewers' comments. We cannot make any decision about publication until we have seen the revised manuscript and your response to the reviewers' comments. Your revised manuscript is also likely to be sent for further evaluation by the reviewers.

We expect to receive your revised manuscript within 3 months. 

**IMPORTANT - SUBMITTING YOUR REVISION**

Your revisions should address the specific points made by each reviewer. Having discussed these comments with the Academic Editor, we think you should address the reviewers' concerns with more data where possible. However, we recognize that the requests of reviewer 3 are probably beyond what can reasonably be achieved in a normal revision period. Please also assess carefully which of the reviewers' concerns might be addressed by softening conclusions or discussing alternative interpretations of the data. In these cases, please provide a clear justification in the rebuttal on why the requested experiments are not essential for the main message of the paper. 

Please submit the following files along with your revised manuscript:

*Re-submission Checklist*

*Published Peer Review*

*PLOS Data Policy*

*Blot and Gel Data Policy*

Sincerely,

Gabriel Gasque

Senior Editor

on behalf of

Ines Alvarez-Garcia, PhD,

Senior Editor,

djiang@plos.org,

PLOS Biology

REVIEWS:

Reviewer #1: This is a well-executed and interesting study by Ivanovitch and colleages combining genetic lineage labelling with single cell sequencing of cardiac progenitors in the mouse. They performed an extensive analysis of genetically labelled cardiac progenitors emerging from the primitive streak at different times. To determine the molecular signatures of these cells, they perform single cell sequencing. High resolution clustering identified subpopulations of the FHF, AHF and SHF. Their data suggests that cardiac progenitors are already molecularly distinct as they gastrulate, which the authors suggest might be due to exposure to spatio-temporally distinct signalling environments.

The paper is well-written and clear with high quality data and figures.

Minor changes:

Figure 3: it would be helpful to indicate with a line the extent of the PS, and the location of the node and proximal/distal PS

The description of Fig. 4 in the text and the figure legend is a bit confusing: (236) We next administered tamoxifen in Foxa2nEGP-CreERT2/+ R26Rtdtomato/+ embryos at E6+21h and fixed the embryos 10 hours later. - perhaps add = E7+7h to make it consistent with the legend, which says: (891) Localisation of the tdtomato+ cell from the Foxa2 lineage are assessed at (i) E7+7h. 

(244) Typo: (yellow arrows in Fig 2Aiv) should be Fig 4Aiv

Figure 5A, additional labelling of embryos and view displayed would help to orientate the reader, panels Aii and Aiv are not described in legend (arrows) and not mentioned in the main text

Also comment on Figure 6D in the main text

Line 278/9, incomplete sentence: Within these subpopulations, cardiac progenitors belonging to the previously defined FHF, AHF and pSHF. 

Reviewer #2: In this manuscript by Ivanovitch et al., the authors interrogate the model that the primitive streak harbors spatially distinct cardiac sublineages pre-determined for the FHF and SHF structures. They cleverly make use of genetic lineage tracing from T and Foxa2 expressing cells to label emerging mesodermal progenitors in order to perform microscopy and scRNA sequencing analysis during gastrulation. The central thesis is that spatiotemporal segregation of these progenitors at early stages underlies later cardiac fate specification, concluding that the left ventricular progenitors are the first to exit the primitive streak, followed second by the right ventricular progenitors, and thirdly the outflow tract and atrial progenitors, which also arise from different regions. Although previous data (Lescroart et al 2014, Devine et al 2014) is consistent with the these findings, the authors do advance knowledge by demonstrating a graphic map of cardiogenesis. The scRNAseq data and analysis, although insufficient to stand on their own, are a suitable complement to the lineage tracing data, and provide some molecular evidence for temporal (though not spatial) segregation of mesoderm progenitors. Revision work should focus on clarifying the authors' scientific reasoning process (there are places in the manuscript where the strength of the stated conclusions is excessive for the level of evidence presented), and on providing better commentary and discussion to address how the findings fit into the larger scientific field. 

Major Points

In figure 1 the authors showcase an ordered contribution of cells exiting the primitive streak into LV, RV, and OFT and atrial progenitors using single pulse labeling with tamoxifen of the T lineage at early post-gastrulation stages of embryonic development. While embryonic labeling and tamoxifen/Cre labeling kinetics were well vetted and the image analyses supports the authors' claims, the authors rely on sparse labeling of LV cells from their earliest timepoint as the primary explanation for why LV progenitors are the first to exit the primitive streak. The authors could address other explanations beyond early exit from the PS for why LV progenitors are not being captured by T lineage tracing to strengthen their claims, and overall some of the lineage mapping arguments should be enhanced or revised, particularly with respect to their strong conclusions about the spatial separation of progenitors during and before gastrulation. 

For example, in Figure 3, the authors examine spatial patterns for expression of T+ and Foxa2+ cells and conclude from whole-mount immunostaining that a previously uncharacterized population of double positive cells exists at the distal tip of the PS, near the node, and in the definitive endoderm. They assert that these are ventricular CM progenitors on the basis of similar timing for tamoxifen lineage labeling between T-CreERT2 and Foxa2-CreERT2. The authors could be more clear here about what is speculative and what is conclusive from their imaging data with regards to claims about ventricular vs. atrial lineage contributions. The graphic model in Figure 7 holds that ventricular and outflow tract CMs derive from anterior primitive streak separately from atrial CMs. However, there are some logical inconsistencies with concluding that ventricular CM progenitors strictly originate from the anterior primitive streak: 1. since only 50% of the cardiac crescent is made up of Foxa2 progeny (Bardot et al 2017), it is conceivable that a proportion of ventricular CM progenitors originates from a region other than the distal primitive streak; 2. in published single clone analysis of mesoderm progenitors (Devine et al 2014), many ventricle-atrium clones are noted, suggesting that ventricular and atrial CM progenitors may originate from similar primitive streak locations; 3. all myocardium is derived from Mesp1-lineage progenitors (Saga et al 2000 and many others), but at the time of birth of LV progenitors ~E6.5, the anterior streak is actually devoid of Mesp1 (see present manuscript Figure 6B cluster aPS, also ISH examples in Cunningham et al 2017 Figure 1, and Saga et al 2000 Figure 1). If there is better evidence for an aPS origin for these structures, particularly derived from Figure 3 or 4, the logic should be described in the manuscript text more explicitly. An alternate (and more modest) explanation is that the RV and LV progenitors emerge from proximal/posterior PS, since posterior epiblast transiently ("accidentally") expresses Foxa2 as the streak extends, and production switches from endoderm to mesoderm (Probst Biorxiv 2020, Figures 2 and 3). 

In figure 2 the authors utilize Foxa2 lineage tracing to investigate origins of ventricular vs. atrial myocardium cells. Interestingly, labeling with Foxa2 at the same early timepoint of T lineage tracing yielded labeling of both LV and RV myocardium (line 198-201) however lineage labeling at the same stage with T did not produce robust labeling of LV cells in figure 1. In both figures the lineages look pretty sparse but the interpretation discrepancy is not elaborated on in the authors' conclusion that Foxa2+ primitive streak cells contribute to the ventricular myocardium at similar stages to T+ cells (line 203-204). 

Similarly with the above, it is not conclusive that atrial progenitors necessarily arise from proximal primitive streak (lines 251-2) based on absence of Foxa2 in that location, since it is conceivable that they arise distally but are just Foxa2 negative. To draw this conclusion definitively requires that all mesoderm derivatives (or at least those destined for the cardiac crescent) from distal PS are also Foxa2-positive or Foxa2-lineage-positive. If the authors use Figure 3 to support this claim, the logic should be described in the manuscript text more explicitly (lines 247-9). It might be helpful to have orthogonal view projections or slices (axial or coronal) to accompany 3A, 3C, or 3E, because this would show the different primitive layers. 

The authors use figure 4 to illustrate the potential for a stream of migratory cells (distal PS to anterior embryo) corresponding to presumptive cardiac crescent. Addition of known marker genes and a more clear distinction between speculation and definitive conclusions could aid in conclusions drawn from this figure. 

In figures 5 and 6 the authors utilize scRNAseq to interrogate molecular signatures of RV, OFT, and atrial cardiac progenitors within their T lineage labeled embryos, and to further support the distinction between progenitor pools. The analysis is curated to support the temporal segregation of progenitors story told in earlier figures with the microscopy data, although the authors could make the complementation of the findings more clear in their text to strengthen their spatial and temporal conclusions shared in this section of the manuscript. The authors also make use of published datasets to understand subpopulations within their own data, and could elaborate more on their methods for doing so. Additionally, more explanation for how cell types were labeled (what marker genes were used?) would benefit reader interpretation of UMAPs, particularly in Fig 5D. Interestingly, as pointed out above, expression of Mesp1 is undetectable in the aPS population relative to other identified regions in fig6B - could the authors explain their interpretation of this finding in relation to referenced published works from Lescroart and Devine wherein Mesp1 lineage is shown to contribute to heart? Additionally, Hand2 and Isl1 in Figure 5E are SHF markers - why are these genes highly detected in the cluster labeled "LV" relative to RV, OFT, and atria? The authors could also provide more methods explanation for the discrepancy between ages of embryos analyzed via microscopy (generally a few hours older) and embryos analyzed via scRNAseq (generally a few hours younger), as well as more notes on the methods and interpretations for performing the cell type and tdTomato lineage labeling in their scRNAseq analysis. The tdTomato expression was compelling in the microscopy images - the authors could comment more on the trends they observe with the lineage marker within the scRNAseq data. It was difficult to determine which conclusions were firm and which were more speculative in this section of the manuscript, especially with regards to the signaling gradients discussed. 

Overall the scRNAseq computational methods should be better explained. For example, the authors draw from prior published scRNAseq datasets, but do not describe any specifics that contributed to their cluster assignment (lines 278-9 contain an incomplete sentence). 

Fig 5C-F is particularly problematic, as it assigns anatomical futures to groups of cells. This is not appropriate and needs to be revised to reflect simply that there are different clusters at E7.5, some of which contain tdTomato.

The use of the tam@E6+5h in Figure 6 isn't well explained, especially since this time point is not represented in Figure 1 - unless detailed further, the differential gene expression could appear to result from embryo vs. extraembryonic mesoderm differences (lines 342-344), not ventricular vs. outflow tract progenitors (lines 357-9, 360-1). The re-clustering process used to generate this figure should be explained better or overtly indicated in the figure. The relationship of tdTomato-expression to the clusters show in 6E-H is needed to help clarify this. The tdTomato lineage is not shown or described in Fig 6, making this figure no more than an incomplete recapitulation/reanalysis of published data, and therefore without a link to the previous data should be removed. As with Figure 5, there is no new information to help understand the spatiotemporal experiments presented with the lineage tracing.

Overall, the data presented is of high quality, and provides a framework for a manuscript that will be impactful. To help this paper convey its message more clearly, the authors should focus on providing better explanations of the methods and observations, in addition to re-writing conclusion statements to better account for alternate possibilities. 

Minor points

How did the authors decide how many embryos were sufficient for their imaging conclusions?

To what depth (reads/cell) were the analyzed scRNAseq libraries sequenced?

Figure s6 is missing panel E as described in the figure legend

Fig 5: showing Cerulean in green and GFP in blue is confusing

Reviewer #3: In this study Ivanovitch and colleagues used combined lineage tracing, via inducible T-CreERT2 and Foxa-CreERT2 and R26R-tdTomato reporter crosses, and scRNA-Seq analyses to fate map and molecularly profile primitive streak (PS) gastrulation stages to investigate the spatiotemporal patterning of cardiac progenitors. The authors propose that cardiac progenitors are prepatterned in the PS and this determines their contribution to distinct anatomical compartments within the forming heart. Specifically, they identified that ventricular and outflow tract (OFT) progenitors originate from the distal PS and atrial progenitors from the proximal PS and that there is a sequential contribution, such that LV progenitors emerge first followed by those of the RV and then those of the poles (OFT and inflow tract; IFT). Single cell RNA-Seq identified these sub-populations to be molecularly distinct groups of progenitors and that discrete transcriptional programmes are established before migration out of the PS. The Foxa2 lineage tracing confirmed a common lineage between the RV and OFT within the pre-defined SHF, and supported previous clonal findings, by Meilhac et al. (2014) and Lescroart et al (2014), that atrial progenitors segregate as an independent progenitor pool. 

This study converges on further understanding of cardiac progenitor specification and commitment in the early mouse embryo, with insight into how discrete progenitor pools and subpopulations can emerge coincidentally to contribute to distinct anatomical regions of the developing heart. However, there are issues with the study in its current form and some over-reach on interpretation that requires further experimental data to support:

General comment:

The conclusions of the study are thus far based exclusively on lineage tracing and SCRNA-Seq. However, there appears to be an over extrapolation from two types of experiments that are only weakly linked by the RNAseq having been performed on labelled samples. Importantly, there is a lack of validation experiments and no genetic proof of the spatiotemporal patterning described; within, and emerging from, the PS. Two suggestions to provide important validation are as follows:

i) the authors should carry out some low-throughput spatial transcriptomics, ie RNA FISH, RNAScope or HCR to localise some of the specific sub-population markers to discrete stages and regions of the PS; multiplexing markers here that have emerged from each of the sub-populations defined by the scRNA-seq would serve as an important validation of the early spatially distinct specification within the PS; 

and/or 

ii) utilising a specific marker(s) emerging from the scRNA-seq data the authors c/should generate a Cre-driver for more refined lineage tracing of the contribution from PS sub-populations into the heart. Alternatively, this could be used for lineage ablation crossing with a DTA-floxed mouse (available from JAX) and phenotyping, or they could target a specific marker (as validated by the FISH/RNAScope/HCR suggested above) with either of the T-CreERT2 or Foxa2-CreERT2 cre-drivers. One or more of these approaches would not only support the identification of discrete cardiac progenitor pool(s) but also the contribution to a distinct anatomical region, and would map onto the predicted insights into specific forms of congenital heart disease. 

Specific comments:

1) The T+ and Foxa2+ lineage tracing studies are carried out exclusively utilising the single R26R-tdTomato reporter line. Whilst this is a widely used reporter, there is an acceptance that any reporter may capture a unique pattern with potential for ectopic labelling and so it would be helpful if the authors included at least some of their fate mapping analyses with an alternative reporter (such as R26R-mTmG or R26R-EYFP) to confirm and reinforce the specific labelling with each of the Cre drivers. This is also important when considering the reliance of tdTomato-expression to discriminate molecular profiles and subpopulations of progenitors in the scRNA-Seq analyses. This is not a request to repeat the study with an alternative reporter, simply to validate some of the initial lineage tracing.

2) The T-lineage labelling experiments (Figure 1) are quite confusing and lack evidence that variations in staging of embryos account for differences in contributions to LV and RV. This c/should be tested by labelling earlier. 

3) Related to the lineage trace experiments, it is not clear that looking at surface labelling is a good proxy for contribution to the heart. This needs to be validated in a few hearts, combining quantification of surface labelling with disaggregation of labelled cells and flow sorting to more accurately quantify and see if they agree.

4) Tamoxifen administration - the experiments on tamoxifen rate of recombination are unclear. It is surprising that recombination via oral gavage occurs after 2 hours given previous studies. Moreover, these experiments where conducted on E9.5 embryos, which does not appear to relate to any timepoints in the study. In addition, the accompanying PCR results are from pooled embryos and should be conducted on individual embryos to see the amount of recombination occurring per embryo and to monitor variation between embryos per litter. 

Time points chosen - Why does the study use E6+21 and E7; arguably a more valid experiment would be at E6 and E6.5? The text is also misleading here in stating labelling was conducted from E6.5, when in fact it started at E6 +21 hours and as such, given the time for recombination, labelling is more likely from E7.25/7.5 when a large proportion of cardiac progenitors will have already left the PS(?). Also given the manuscript shows tamoxifen at E5 labels CMs why have these timepoints not been been included? - Labelling at E6+21 marks LV, RV and atria, if the authors include earlier timepoints, do they find a regimen that only labels ventricles and atria, ie FHF?

5) Following on from the above, a more detailed analysis needs to be conducted on the location of recombined/labelled cells. Given recombination takes place after 2 hours, analysis c/should be conducted 12 hours following tamoxifen administration in order to localize the cells. This also needs to be conducted for the entire litter to obtain a convincing insight into what is being labelled. This is especially important given that the images of T-labelling seem to show it extending almost to the anterior of the embryo (Figure 3c) and it looks like T is strongly expressed in the mesodermal wing as supported by the scRNA-Seq atlas datasets referenced (Pijuan-Sala 2019) which show that T is most strongly expressed in mesoderm subtypes and not the PS between E7 and E7.75. Detailed imaging, like that carried out for Foxa2, would allow the location of recombined cells to be identified and when combined with a mesoderm marker would further provide clarification as to whether these cells reside in the PS or are differentiating mesoderm. 

6) Foxa2-labelling experiments: tamoxifen at E6+21 (Figure 4) resulted in ventricle and outflow labeling, however, the authors conclude Foxa2 is not expressed in the PS when the switch from ventricle to outflow/atria contribution occurs. Given they detect both RV and outflow labelling, this suggests that outflow and ventricle are labelled early and specified later(?).

7) Based on the time course of tamoxifen administration the authors make the statement that atrial precursors are the last cardiomyocyte precursors to leave the PS. This is not strongly supported by the data given at E6+21 they observe atrial labeling, ie just because they are amongst the last progenitors to be restricted does not necessarily mean atrial precursors leave the streak and are specified earlier. Given clonal experiments have previously shown contributions to parts of the atria and LV are FHF-derived, how do the results herein fit with these prior studies?

8) In a number of places important experimental details are lacking:

i) Whilst the text states 3494 single cell transcriptomes were obtained (page 14), what is the specific number used in this manuscript? i.e. how many FHF, pAHF, AHF cells were included, it appears less than 3494 in S4Fig. 4A, B? 

ii) Not enough details are provided in the text or legend to interpret Figure 5C and the tdTomato contributions to specific clusters.

iii) The single cell RNA-seq analysis requires extended methodology- were the entire embryos dissociated? How many 10x runs were completed per group? Were each group included on the same 10x run? Were there any batch effects and/or differences between groups? What was the rationale for further sub clustering cells as stated in the text? Single cell analysis was done with a labelling regime not previously described (Collected at E6+21 but given tamoxifen at E6+5 (16 hours of labelling) and then collected at E7+3 but given at E7+3). How do these labelling regimes fit with the above lineage analysis conducted at different times?

iv) Detailed and specific details should be provided about the transcriptional state of identified precursor types - currently the data is not clear? How many of each cell type are identified? What is the complete unbiased list of genes which mark these clusters? Which genes overlap, and which do not?

9) Several statements are not supported by the data and especially given the relatively low number of embryos/hearts studied. For example,

i) Lines 200-202: " In 3/14 hearts, smaller tdtomato-expressing domains formed either in the left or right ventricular myocardium. This indicates that independent groups of Foxa2 expressing progenitors exist in the primitive streak that contribute to the left and right myocardium". This cannot be concluded on the basis of 3 hearts. especially if the labelled domains are small. This indicates that labelling might have been clonal or sparse, in which case it is likely to manifest in one or the other (but not both) ventricles, by chance, rather than because of different progenitors for LV and RV.

ii) In several places the authors make inferences on dynamic behaviour eg "...forming a stream of migratory cells originating…", however this is on the basis of fixed samples and as stated above to make such claims the authors require time-lapse data on cultured embryos/hearts.

Iii) Lines 291-294 : "We have summarised these results as a fate map for the different cardiac regions (Fig 7A-B) and we propose that right ventricle, outflow and atrial progenitors form molecularly distinct populations within the AHF and pSHF mirroring their distinct origin in the primitive streak." This claim is an over-interpretation; by definition, a fate mapping experiment is prospective. Much of the inferences in the accompanying figure are predicated on RNAseq and as such the authors have conflated the 'state' of a cell with its 'fate'. Again, this comes back to the main point above around validating the RNAseq data as relates to labelling of subpopulations in the PS, more refined lineage tracing and genetic targeting.

10) The model in Figure 7 is surprisingly detailed given that there is no direct evidence for much of it in the preceding figures. The model shows relative positions of transcriptional sub-types, but there is no direct experimental evidence looking at their location. it also shows arrows indicating movement of cells, and again, there are no data presented to support the cell movements proposed. The suggested validation of the sub-populations within the PS (point 1) above) to include more refined lineage tracing, arguably also requires time-lapse experiments on cultured embryos/ hearts to capture patterns of migration and contributions to the forming heart to support the model proposed. 

Minor points:

- Brightfield images of embryos in Figure 1H are not very clear and do not adequately allow staging to be assessed. Given the use of the T-GFP reporter, images of T expression would also be helpful here.

- Additional clarification around the precise staging is required: what does a late streak or mid-streak embryo refer to, e.g. E6+21, E7, E7+7, E7+11? How do distal and proximal PS relate to anterior PS and timeframes when cells enter the streak?

- Figure 4A - the masked image is confusing and it looks like only the td-tomato image is masked, why? Panel iii is not convincing given there appear to be penetration issues as DAPI is not present, what about the GFP signal?

- Figure 5B shows E7+14 embryos but this is not discussed in the text or legend.

---

## [Decision Letter · Decision Letter 2]

15 Mar 2021

Dear Dr Ivanovitch,

Thank you for submitting your revised Research Article entitled "Ventricular, atrial and outflow tract heart progenitors arise from spatially and molecularly distinct regions of the primitive streak." for publication in PLOS Biology. I have now obtained advice from two of the original reviewers and have discussed their comments with the Academic Editor. 

The reviews are attached below. You will see that while Reviewer 2 is now satisfied, Reviewer 3 still raises several concerns. However, after discussing the reviews with the Academic Editor, we have decided that we will probably accept this manuscript for publication, provided you satisfactorily address the data and other policy-related requests noted below. We think that the conclusions of the manuscript are based on a reasonable and appropriately strong body of evidence.

Please also take this last chance to review your reference list to ensure that it is complete and correct. If you have cited papers that have been retracted, please include the rationale for doing so in the manuscript text, or remove these references and replace them with relevant current references. Any changes to the reference list should be mentioned in the cover letter that accompanies your revised manuscript.

We expect to receive your revised manuscript within two weeks. 

-  a cover letter that should detail your responses to any editorial requests.

*Published Peer Review History*

*Early Version*

Sincerely,

Ines

--

Ines Alvarez-Garcia, PhD,

Senior Editor,

PLOS Biology

Thank you for providing the individual numerical values that underlie the summary data displayed in the figures to comply with our data policy. Nevertheless, we are missing the data of some of the figure panels and we would like you to clarify/address a few points:

Fig. 9E, G, H; Fig. S12B, D; Fig. 13B and Fig. 15A, B

- Some of the columns in Fig. 5E and 5F have exactly the same values, but are labelled differently. Are they supposed to be the same (if so, please explain) or is this a mistake?

- In S9 source data, there are two tabs named Fig. 6B and Fig. 6C, but they seem to belong to Fig. 9B and C – is this correct? If so, please use the correct labels for the tabs and the data titles. Otherwise, please add the missing data for Fig. 9C.

- Please also ensure that figure legends in your manuscript include information on WHERE THE UNDERLYING DATA CAN BE FOUND.

- In your Data Availability Statement you mention that the single cell RNA sequencing data have been deposited in NCBI under the accession number GSE153789. Please make sure that this data is made publicly available before the manuscript is accepted for production.

Reviewer's comments

Rev. 2: Benoit Bruneau, Martin Dominguez, and Alexis Leigh Krup - note that these reviewers have signed their report.

The authors have carefully addressed all of our comments, and considerably improved the work. It should be accepted immediately.

Rev. 3:

This remains an interesting study with a significant amount of data and, in revision, the authors have gone some way to addressing previous concerns both with additional clarifications in the text and new experimental data. However, the fundamental issue remains that they fall short of demonstrating the precise time-frame of cardiac progenitor specification/patterning relative to the primitive streak, and the combined approaches presented remain correlative rather than serve as definitive proof. 

Some general comments that relate to this main issue below:

1. Cardiac progenitors are transcriptionally pre-patterned within the streak - The data provided still does not address or allow this point to be concluded. Whilst there is now increased discussion around the topic the overall conclusion,s as stated in the title and abstract, are not fully supported. Are cells transcriptional pre-patterned in the streak or are they patterned once cells have migrated from the streak? Unfortunately, the data does not provide sufficient insight here. Given, Figure 8 presents later staged embryos (Tam E7+7h - Collected E7+14), it is not surprising that there are transcriptional differences between tdTomato positive and negative cells (based on our current knowledge of heart development). To conclude that cells are transcriptionally prepatterned in the streak would require the identification of these clusters within the streak at earlier stages (see point 2. below about earlier scRNA-seq analysis).

2. scRNA correlation with lineage tracing - "using a single cell transcriptomic assay in combination with genetic lineage tracing" - whilst these two experimental approaches have been conducted they are not combined. One crucial scRNA-seq experiment (Figure 9) that used genetic lineage labelling appears to have been unsuccessful: the authors state they are unable to detect the tomato transcript. This experiment would have allowed some insight into the transcriptional profiles of cells at different maturities within the MS-OB stage embryos, which is when the authors allude to the pre-patterned progenitors, i.e. how does the transcriptional profiles of cells change with varying level of tdTomato transcript?

3. Over-reach in interpreting the genetic lineage-tracing experiments- Whilst the authors have suggested that variations in staging observed within litters (E6+21h litter= ES,MS,LS) reflect LV labelling contribution, this is not appropriately addressed when making other conclusions and in the rebuttal; especially regarding the Foxa2 lineage experiments. Given the transient nature of Foxa2 expression and its expression in the epiblast at early streak stages, an analysis of a full litter is needed to accurately assess the cell labelling dynamics; in contrast to the author's rebuttal. Evidence that RV and OFT progenitors are specified in the streak could also be interpreted as occurring once they leave the streak. The model in Figure 11 shows the RV is specified before OFT, however, there is an embryo presented which has RV labelling alone (no OFT) when tamoxifen administered at E7+7 which could suggest that RV progenitors are still emerging from the streak. "independent groups of Foxa2 expressing progenitors exist in the primitive streak that contribute to the left and right myocardium" - in this case, 10/14 hearts have both LV and RV labelling, the 3/14 embryos used to support this conclusion could in fact reflect staging differences which will in turn influence the extent of labelling i.e. an embryo with labelling exclusively in the LV of the heart may be less mature relative to those that reveal labelling within the RV/OFT hearts. 

In summary, whilst this paper has a significant amount of data, which overall is nicely presented, the multiple different genetic and experimental approaches are correlative and all have caveats which prevent strong conclusions on the precise timing of progenitor patterning. Overall from the evidence presented it is difficult to reconcile that cardiac progenitors are specified in the primitive streak. Being able to separate timing of emergence/anatomical location and spatio-temporal pre-specification of progenitors is very challenging, but unfortunately not adequately addressed herein.

If the authors are unwilling to go further to prove the timing of specification/patterning they should tone down their conclusions significantly which includes amending the title of the study and the abstract.

---

## [Editor Report · Decision Letter 3]

23 Mar 2021

Dear Dr Ivanovitch,

On behalf of my colleagues and the Academic Editor, Sally Lowell, I am pleased to say that we can in principle offer to publish your Research Article entitled "Ventricular, atrial and outflow tract heart progenitors arise from spatially and molecularly distinct regions of the primitive streak" in PLOS Biology, provided you address any remaining formatting and reporting issues. These will be detailed in an email that will follow this letter and that you will usually receive within 2-3 business days, during which time no action is required from you. Please note that we will not be able to formally accept your manuscript and schedule it for publication until you have made the required changes.

PRESS

Thank you again for supporting Open Access publishing. We look forward to publishing your paper in PLOS Biology. 

Sincerely, 

Ines

--

Ines Alvarez-Garcia, PhD 

Senior Editor 

PLOS Biology